# rbFOX1/MBNL1 competition for CCUG RNA repeats binding contributes to myotonic dystrophy type 1/type 2 differences

Chantal Sellier[1], Estefanía Cerro-Herreros[2,3], Markus Blatter[4], Fernande Freyermuth[1], Angeline Gaucherot[1], Frank Ruffenach[1], Partha Sarkar[5], Jack Puymirat[6], Bjarne Udd[7,8,9], John W. Day[10], Giovanni Meola[11,12], Guillaume Bassez[13], Harutoshi Fujimura[14], Masanori P. Takahashi[15], Benedikt Schoser[16], Denis Furling[13], Ruben Artero [2,3], Frédéric H.T. Allain[4], Beatriz Llamusi[2,3] & Nicolas Charlet-Berguerand[1,17,18,19]

Myotonic dystrophy type 1 and type 2 (DM1, DM2) are caused by expansions of CTG and CCTG repeats, respectively. RNAs containing expanded CUG or CCUG repeats interfere with the metabolism of other RNAs through titration of the Muscleblind-like (MBNL) RNA binding proteins. DM2 follows a more favorable clinical course than DM1, suggesting that specific modifiers may modulate DM severity. Here, we report that the rbFOX1 RNA binding protein binds to expanded CCUG RNA repeats, but not to expanded CUG RNA repeats. Interestingly, rbFOX1 competes with MBNL1 for binding to CCUG expanded repeats and overexpression of rbFOX1 partly releases MBNL1 from sequestration within CCUG RNA foci in DM2 muscle cells. Furthermore, expression of rbFOX1 corrects alternative splicing alterations and rescues muscle atrophy, climbing and flying defects caused by expression of expanded CCUG repeats in a *Drosophila* model of DM2.

[1] IGBMC, INSERM U964, CNRS UMR7104, University of Strasbourg, 67404 Illkirch, France. [2] Translational Genomics Group, Interdisciplinary Research Structure for Biotechnology and Biomedicine BIOTECMED, University of Valencia, 46010 Valencia, Spain. [3] INCLIVA Health Research Institute, 46010 Valencia, Spain. [4] Institute for Molecular Biology and Biophysics, Swiss Federal Institute of Technology (ETH) Zurich, 8092 Zurich, Switzerland. [5] Department of Neurology, University of Texas Medical Branch, Galveston, TX 77555, USA. [6] Human Genetics Research Unit, Laval University, CHUQ, Ste-Foy, Quebec QC G1V 4G2, Canada. [7] Neuromuscular Research Center, Tampere University Hospital, 33521 Tampere, Finland. [8] Department of Medical Genetics, Folkhälsan Institute of Genetics, Helsinki University, 00290 Helsinki, Finland. [9] Department of Neurology, Vasa Central Hospital, 65130 Vaasa, Finland. [10] Department of Neurology, Stanford University, San Francisco, CA 94305, USA. [11] Department of Biomedical Sciences for Health, University of Milan, 20097 Milan, Italy. [12] Neurology Unit, IRCCS Policlinico San Donato, San Donato Milanese, 20097 Milan, Italy. [13] Sorbonne Université, Inserm, Association Institut de Myologie, Center of Research in Myology, 75013 Paris, France. [14] Department of Neurology, Toneyama National Hospital, Toyonaka 560-0045, Japan. [15] Department of Neurology, Osaka University Graduate School of Medicine, Suita 565-0871, Japan. [16] Friedrich-Baur-Institute, Department of Neurology, Ludwig Maximilian University, 80539 Munich, Germany. [17] UMR7104, Centre National de la Recherche Scientifique, 67404 Illkirch, France. [18] Institut National de la Santé et de la Recherche Médicale, U964, 67404 Illkirch, France. [19] Université de Strasbourg, 67404 Illkirch, France. Correspondence and requests for materials should be addressed to B.L. (email: Mbeatriz.Llamusi@uv.es) or to N.C.-B. (email: ncharlet@igbmc.fr)

Myotonic dystrophy is the most common muscular dystrophy in adults and comprises two genetically distinct forms. Myotonic dystrophy type 1 (DM1) and its severe congenital form (CDM1) are caused by an expansion of CTG repeats in the 3′-untranslated region (UTR) of the *DMPK* gene[1–3]. In contrast, myotonic dystrophy type 2 (DM2) is caused by an expansion of CCTG repeats within the first intron of the *CNBP* (also known as *ZNF9*) gene[4]. Expression of mutant RNAs containing hundreds to thousands of CUG or CCUG repeats interferes with the metabolism of other RNAs through dysfunctions of mainly two classes of RNA binding proteins. First, expression and phosphorylation of the CUG-binding protein 1 (CUGBP1, encoded by the *CELF1* gene) are increased in DM1 heart samples, especially in the most severely affected individuals[5]. Second, Muscleblind-like proteins (MBNL1, MBNL2 and MBNL3), which are RNA binding proteins specifically recognizing YGC RNA motifs, are titrated away from their normal mRNA targets as a result of their binding to expanded CUG and CCUG RNA repeats. MBNL proteins titration is illustrated by their mislocalization within nuclear RNA foci formed by expanded CUG and CCUG repeats in cell and animal models of myotonic dystrophy[6–8]. MBNL and CUGBP1 are RNA binding proteins that regulate pre-mRNA alternative splicing[9–14]. Thus, alterations of MBNL and CUGBP1 functional levels result in reversion to embryonic splicing patterns for various mRNAs, which are associated with several symptoms of myotonic dystrophy[15–22]. Importantly, knockout of Mbnl1 and/ or Mbnl2 in mouse reproduces splicing alterations and keys features of myotonic dystrophy[10,13,14,20,23]. Conversely, overexpression of MBNL1 corrects splicing alterations and myotonia in mice expressing expanded CUG repeats[24]. Finally, a higher number of pathogenic CUG repeats leads to a greater titration of MBNL proteins, resulting in increased RNA metabolism alterations that correlate with increased disease severity in DM1 individuals[23,25]. These results highlight the importance of MBNL titration in myotonic dystrophy type 1.

Myotonic dystrophy type 2 resembles adult-onset DM1 with autosomal dominant inheritance pattern and similar clinical multi-organ features, including progressive skeletal muscle atrophy and weakness, myotonia, cardiac arrhythmia and conduction defects, posterior subcapsular iridescent cataract, insulin resistance, hypogammaglobulinemia, as well as cognitive and personality changes[26–28]. Despite these similarities, there are significant differences between DM1 and DM2, including a usually more favorable clinical course in DM2 compared to DM1. Indeed, symptoms are generally milder in DM2 compared to DM1 and include slower and less severe progression of the disease, reduced severity of the cardiac involvement, later and less prominent weakness of the respiratory, facial and bulbar muscles, less evocable myotonia and preserved social and cognitive abilities[29–31]. A milder involvement in DM2 compared to DM1 is also found at the cellular level as in vitro cultures of muscle cells originating from individuals with DM1 reveal reduced fusion capacity and alternative splicing alterations compared to control cells. However, only subtle or even no detectable fusion defects or splicing alterations are observed in cultures of DM2 myoblasts[32–34]. Paradoxically, these milder clinical and cellular features are in contradiction with the 3 to 5 folds higher expression of the *CNBP* mRNA, which first intron hosts the expanded CCUG repeat, compared to the *DMPK* mRNA, which 3′UTR hosts the expanded CUG repeats[7,35,36]. Furthermore, the average size of expanded repeats is generally higher in DM2 (up to ~11,000 CCTG repeats in blood) compared to adult-onset DM1 (up to ~1000 CTG repeats in blood)[4]. Thus, expanded CCUG repeats appear inherently less pathogenic than expanded CUG repeats. A likely contribution to this difference of toxicity is their genomic localization as CCUG repeats are embedded in the first intron of the *CNBP* pre-mRNA, which is presumably less stable compared to the 3′UTR of the *DMPK* mRNA that hosts the CUG repeats. In favor of this hypothesis, transgenic *Drosophila* expressing either expanded CUG or CCUG repeats, embedded in a comparable genomic context deprived of any *DMPK* or *CNBP* sequences, show similar DM-like phenotypes[37,38]. Nonetheless, it is not excluded that other mechanisms may further contribute to the lesser toxicity of expanded CCUG repeats in DM2.

Searching for novel RNA binding proteins that interact specifically with either CUG or CCUG expanded repeats, we identified that the rbFOX RNA binding proteins bind to expanded CCUG repeats, but not to expanded CUG repeats. The rbFOX family comprises three members, rbFOX1 (also named Fox-1, A2BP1 or HRNBP1), rbFOX2 (also known as Fox-2, RBM9, Fxh or HRNBP2) and rbFOX3 (also called Fox-3, NeuN or HRNBP3), which are involved in the regulation of various aspects of mRNA metabolism[39–47]. While rbFOX2 is widely expressed, rbFOX1 is enriched in skeletal muscle, heart and brain, and rbFOX3 expression is restricted to neuronal cells[48]. All three rbFOX proteins possess a near-identical RNA-recognition motif (RRM) that recognizes the UGCAUGY RNA sequence[39–41,49].

Here, we find that rbFOX1 also binds to expanded CCUG repeats, albeit with a lesser affinity compared to its favorite UGCAUGY sequence. Interestingly, rbFOX1 co-localizes with CCUG RNA foci in muscle cells and skeletal muscle tissues of individuals with DM2. In contrast, rbFOX1 does not bind to expanded CUG repeats and does not co-localize with CUG RNA foci in DM1 samples. Furthermore, we find that the binding of MBNL1 and rbFOX1 to expanded CCUG repeats are mutually exclusive. Addition of rbFOX1 competes with MBNL1 binding to CCUG repeats and partially releases MBNL1 from CCUG RNA foci in DM2 muscle cells. Importantly, expression of rbFOX1 corrects splicing alterations and reduces muscle atrophy, as well as climbing and flying defects caused by expression of expanded CCUG repeats in *Drosophila* models of DM2. Thus, we propose that the rbFOX proteins, by competing with and reducing the titration of MBNL1 within CCUG RNA foci, may participate to the lesser toxicity of the CCTG repeat expansion in myotonic dystrophy type 2.

## Results

**Identification of proteins associated with expanded CCUG repeats**. To identify novel RNA binding proteins involved in myotonic dystrophy, we incubated radioactively labeled CUG or CCUG RNA repeats with nuclear extract of differentiated mouse C2C12 muscle cells and analyzed the RNA-bound proteins by UV-light crosslink followed by SDS page gel electrophoresis (Fig. 1a). Consistent with the pioneering work of the Swanson group[6], a UV-crosslink signal around 42 to 45 kDa, which corresponds to Mbnl1 molecular weight, was evident for both CUG and CCUG RNAs. In contrast, a signal at 35 to 40 kDa was observed only with CCUG RNA repeats (Fig. 1a). To identify this factor, proteins from differentiated C2C12 muscle cells were captured on streptavidin resin coupled to biotinylated RNA composed of thirty CUG or CCUG repeats, eluted, separated on SDS–PAGE gels, silver stained and each protein band was cut, gel extracted and proteins were identified by nanoLC/MS-MS analysis (Supplementary Table 1). Mass spectrometry revealed that the band of ~35 to 40 kDa contains various proteins, including Tra2a (33 kDa), Pcbp1 and Pcbp2 (37 and 38 kDa), Hnrnpa3 (40 kDa), Tiar (43 kDa) and rbFox1 (38 kDa). Importantly, only rbFox1 was preferentially associated with CCUG repeats and not with CUG repeats (Supplementary Table 1). We repeated these capture experiments using mouse brain nuclear extract. Silver

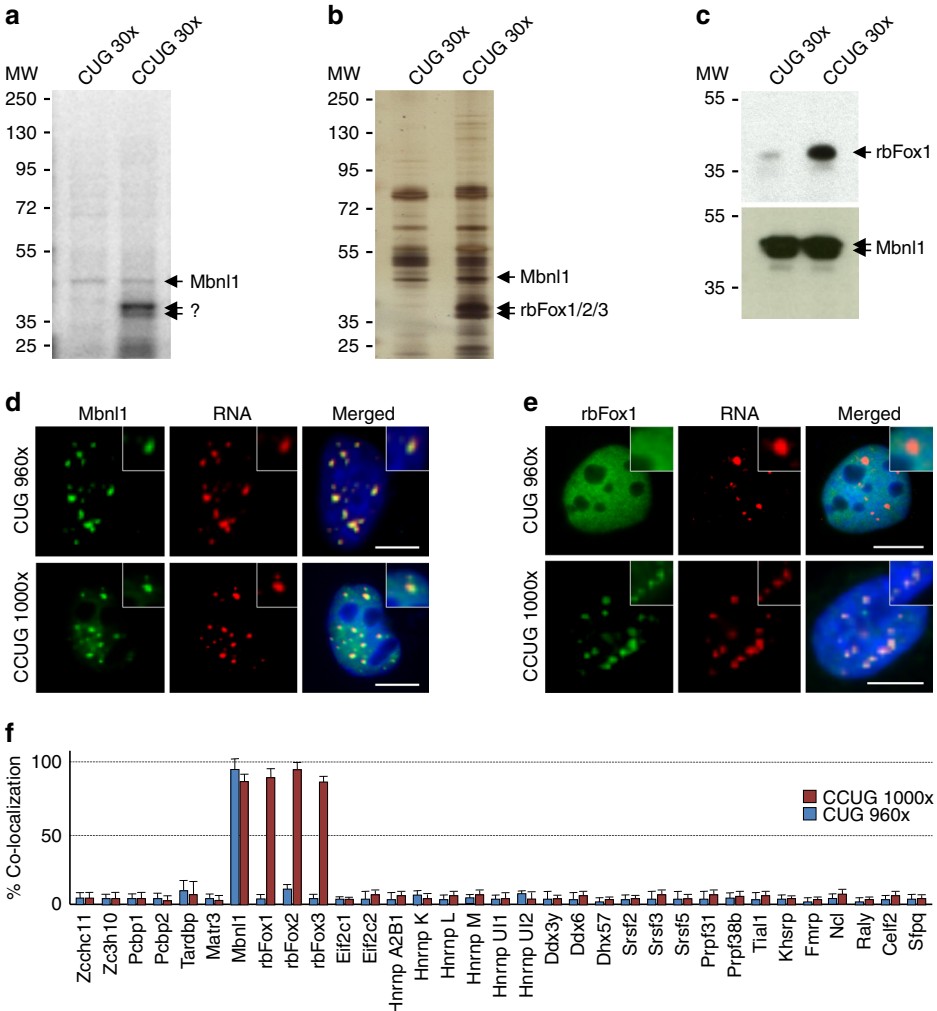

**Fig. 1** Identification of proteins specifically associated with expanded CCUG repeats. **a** UV-crosslinking binding assays of 20 μg of nuclear extract from C2C12 muscle cells differentiated four days incubated with 30,000 CPM of uniformly [αP$^{32}$] internally labeled in vitro transcribed RNAs containing 30 CUG or CCUG repeats. **b** Silver staining of proteins extracted from 1 mg of mouse brain and captured on streptavidin resin coupled to biotinylated RNA containing 30 CUG or CCUG repeats. **c** Western blotting against either rbFox1 or Mbnl1 on mouse brain proteins captured by RNA-column containing either 30 CUG or 30 CCUG repeats. **d** RNA FISH against CCUG repeats coupled to immunofluorescence against Mbnl1 on differentiated C2C12 cells transfected with a plasmid expressing either 960 CUG or 1000 CCUG repeats. **e** RNA FISH against CCUG repeats coupled to immunofluorescence against rbFox1 on differentiated C2C12 cells transfected with a plasmid expressing either 960 CUG or 1000 CCUG repeats. Scale bars, 10 μm. Nuclei were counterstained with DAPI. **f** Quantification of the co-localization of CUG or CCUG RNA foci with candidate proteins in transfected C2C12 cells. Error bars indicate s.e.m. of three independent experiments. Representative images are presented in Supplementary Fig. 1

staining of the captured proteins confirmed that proteins of ~35 to 40 kDa were specifically pulled-down by the CCUG RNA repeats (Fig. 1b). Gel extraction followed by mass spectrometry analysis identified these proteins as rbFox1, rbFox2 and rbFox3 (Supplementary Table 2). Western blotting analysis of the proteins eluted from the RNA affinity columns confirmed that rbFox1 binds to CCUG repeats, but not to CUG repeats (Fig. 1c). As a positive control, Mbnl1 binds to both CUG and CCUG repeats (Fig. 1c).

To confirm these results, candidate proteins identified by mass spectrometry were tested for co-localization with RNA foci of either expanded CUG or CCUG RNA repeats in transfected muscle C2C12 cells. As previously reported, Mbnl1 co-localizes with RNA foci of both expanded CUG and CCUG repeats (Fig. 1d). In contrast, rbFox1, rbFox2, and rbFox3 co-localize only with RNA foci of expanded CCUG repeats, but not with foci of expanded CUG repeats (Fig. 1e and Supplementary Fig. 1). All other tested candidate proteins do not co-localize significantly

with CUG or CCUG RNA foci and thus were not investigated further (Fig. 1f and Supplementary Fig. 1).

**rbFOX1 directly binds to expanded CCUG repeats RNA.** Presence of rbFOX1 within a large protein complex[46] questions whether rbFOX1 contacts directly CCUG RNA repeats or requires indirect protein-protein interactions. Both gel-shift and UV-crosslinking experiments demonstrated that purified recombinant GST-tagged rbFOX1 directly binds to CCUG repeats, but not to CUG repeats (Fig. 2a and Supplementary Fig. 2A, B). Similarly, rbFOX2, which contains a RNA-recognition motif identical to rbFOX1, also binds to CCUG repeats, but not to CUG repeats (Supplementary Fig. 2C). As a positive control, MBNL1 binds to both CUG and CCUG repeats (Fig. 2b and supplementary Fig. 2D). Of interest, MBNL1 binds to CUG or CCUG repeats with higher affinity compared to its natural *BIN1* or *INSR* pre-mRNA targets (Supplementary Fig. 2E). In contrast, rbFOX1

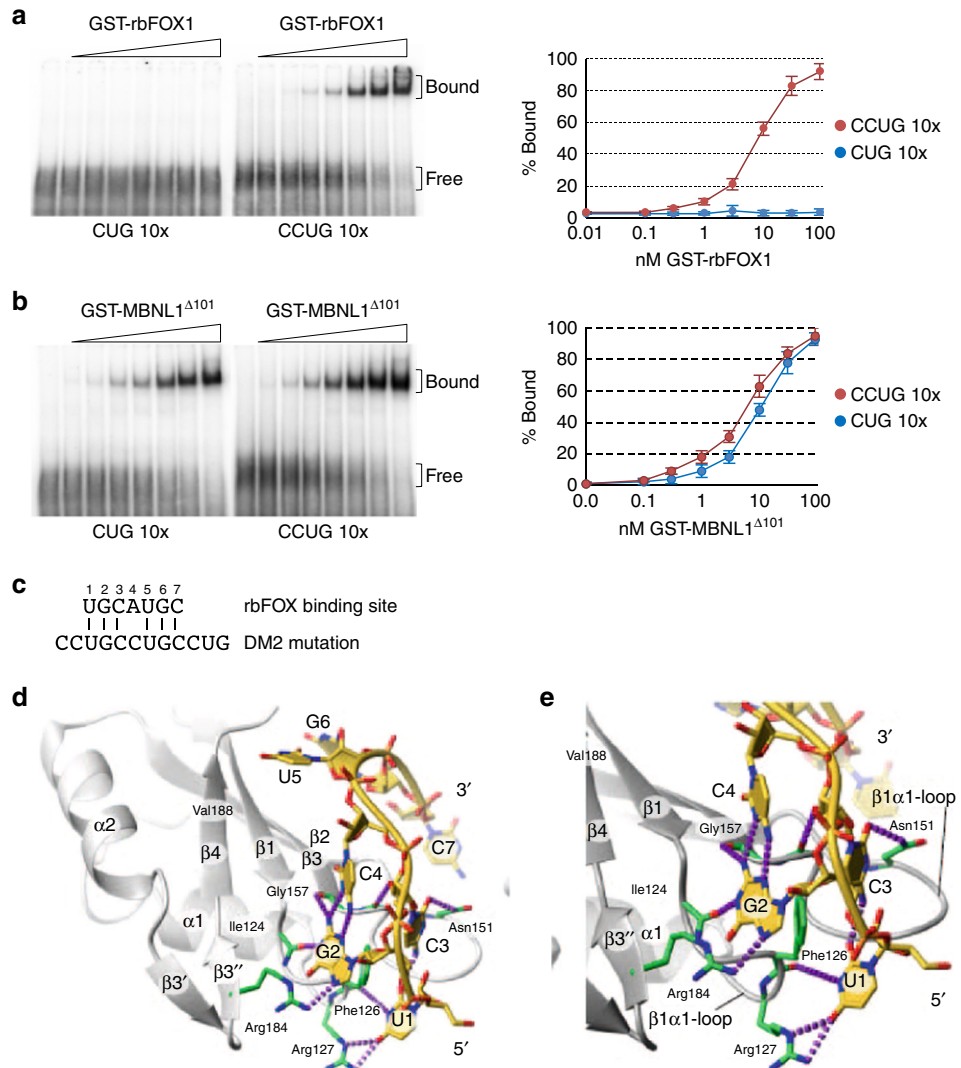

**Fig. 2** rbFOX1 binds to expanded CCUG RNA repeats. **a** Left panel, gel-shift assays of 0, 0.1, 0.3, 1, 3, 10, 30, and 100 nM of purified recombinant GST-rbFOX1 with 10 pM (3000 CPM) of uniformly [αP$^{32}$] internally labeled in vitro transcribed RNAs containing either 10 CUG or CCUG repeats. Right panel, gel-shift quantification. **b** Gel-shift as in **a** but with purified recombinant GST-MBNL1$^{\Delta101}$. **c** Alignment of the UGCAUGC consensus RNA binding site for rbFOX1 with expanded CCUG repeats that constitute the DM2 mutation. **d** Model of rbFOX1 RRM bound to UGCCUGC. RNA is shown in stick representation (yellow) and potential hydrogen bonds in dashed lines (purple). **e** Magnification of **d** showing that guanine 2 and cytosine 4 form a non-Watson-Crick base pair, in an analogous way to guanine 2 and adenine 4 in rbFOX1 RRM bound to UGCAUGU described previously (pdb 2ERR)[49]. Error bars indicate s.e.m. of five independent experiments

binds to ten CCUG repeats with a 2 to 5 folds lesser affinity compared to its known UGCAUGC RNA motif or to its natural *BIN1* or *INSR* pre-mRNA targets (Supplementary Fig. 2F). Competition experiments confirmed that rbFOX1 binds preferentially to the UGCAUGC RNA motif compared to CCUG repeats (Supplementary Fig. 2G). Of interest, binding of rbFOX1 to two CCUG repeats (UGCCUGCC RNA sequence) is negligible compared to the UGCAUGC sequence (Supplementary Fig. 2F), indicating that significant binding of rbFOX1 to the CCUG RNA motif required repetition of that motif. Apparent $K_D$ are indicated in the supplementary Fig. 2H.

Next, we investigated how rbFOX1 can interact, even with a weak affinity, with expanded CCUG RNA repeats. Previously determined NMR structure of the RNA-recognition motif (RRM) of rbFOX1 in complex with the UGCAUGU sequence shows that the central adenosine (underlined) is not involved in any direct RNA-protein interaction[49]. Strikingly, replacement of this adenine by a cytosine yields a motif matching perfectly DM2

mutation (Fig. 2c), questioning whether rbFOX1 can accommodate a cytosine instead of the adenine in the UGCAUGC RNA motif. Indeed, energy minimization and computer modeling of rbFox1 RRM in complex with UGCCUGC RNA shows that the substitution to a cytosine in position four allows the base to adopt a similar base pairing with guanine two compared to the previously shown adenine-guanine interactions (Fig. 2, e). Importantly, we confirmed this model by NMR spectroscopy, and comparison of the amide backbone chemical shifts of both complexes (Supplementary Fig. 2I, 2J) indicates that the number of intra RNA hydrogen bonds is preserved and supports the overall similar binding topology between UGCCUGC and UGCAUGC RNAs. These results confirm that rbFOX1 directly binds to CCUG RNA repeats.

**rbFOX1 co-localizes with CCUG RNA foci in DM2 patients.** We next assessed the localization of endogenous rbFOX proteins in muscle cells and tissues of individuals with DM2. Importantly,

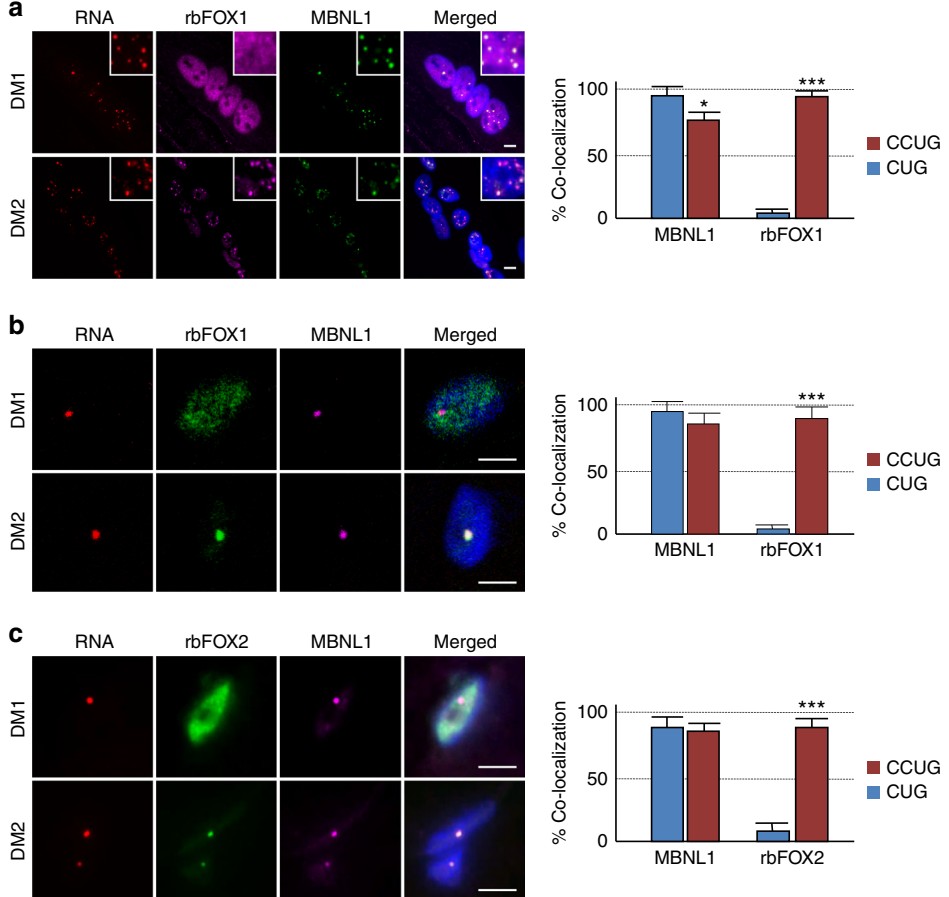

**Fig. 3** rbFOX1 co-localizes with CCUG RNA foci in DM2 patients. **a** Left panel, representative confocal images of RNA FISH against CCUG repeats coupled to immunofluorescence against MBNL1 and rbFOX1 on four days differentiated muscle cells originating from muscle biopsies of individuals with either DM1 or DM2. Right panel, quantification of the co-localization of MBNL1 and rbFOX1 with CUG or CCUG RNA foci. **b** Left panel, CUG or CCUG repeats RNA FISH combined with immunofluorescence against MBNL1 and rbFOX1 on skeletal muscle sections of adult individuals with DM1 or DM2. Right panel, quantification of the co-localization of rbFOX1 within a hundred CUG or CCUG RNA foci. **c** RNA FISH/ immunofluorescence as in **b** but with an antibody directed against rbFOX2. Scale bars, 10 μm. Nuclei were counterstained with DAPI. Error bars indicate s.e.m. of three independent experiments. Student's t-test, asterisk (*) indicates $p < 0.5$, asterisk (***) indicates $p < 0.001$

RNA FISH coupled to immunofluorescence analysis revealed that endogenous rbFOX1 co-localizes with endogenous CCUG RNA foci in primary cultures of differentiated muscle cells originating from skeletal muscle biopsies of individuals with DM2 (Fig. 3a and Supplementary Fig. 3A, 3B). In contrast, rbFOX1 is not present within CUG RNA foci in DM1 muscle cells (Fig. 3a). Concomitant labeling of endogenous MBNL1 demonstrated co-localization of MBNL1 with both CUG and CCUG RNA foci (Fig. 3a and Supplementary Fig. 3B). We confirmed these results in skeletal muscle sections of individuals with myotonic dystrophy. RNA FISH coupled to immunofluorescence indicated that both rbFOX1 and rbFOX2 co-localize with CCUG RNA foci in DM2 muscle sections, but not with CUG RNA foci in DM1 muscle sections (Fig. 3b, c). Concomitant labeling of endogenous MBNL1 demonstrated co-localization of MBNL1 with both CUG and CCUG RNA foci (Fig. 3b, c). As controls, RNA FISH coupled to immunofluorescence analysis indicated that neither endogenous rbFox1 nor rbFox2 co-localize with RNA foci of overexpressed expanded CUG, CGG or AUUCU repeats, which are involved in DM1, Fragile X Tremor and Ataxia Syndrome (FXTAS) and spinocerebellar ataxia of type 10 (SCA10), respectively (Supplementary Fig. 3C and 3D). Overall, these data indicate that rbFOX1 and rbFOX2 are specific components of CCUG RNA foci in myotonic dystrophy type 2.

**rbFOX splicing regulatory functions are not altered in DM2.** To test a potential titration of rbFOX1 within CCUG RNA foci, we first assessed its mobility. Dendra2-tagged rbFOX1 was co-transfected with a vector expressing either no repeats or expanded CUG or CCUG repeats, then a nuclear spot of Dendra2-rbFOX1 was photoconverted from green to red and imaged every minute to follow the nuclear diffusion of rbFOX1. Interestingly, photoconverted Dendra2-rbFOX1 present in nuclear foci of CCUG expressing cells moved away from the CCUG foci and diffused freely in the nucleoplasm, albeit with a slightly slower kinetic compared to rbFOX1 in control condition or with expanded CUG repeats (Fig. 4a). As a control, photoconverted Dendra2-MBNL1 diffuse freely in control cells but was immobilized in foci of expanded CUG or CCUG repeats (Fig. 4b). These data suggest that in contrast to MBNL1, rbFOX1 is not immobilized within CCUG RNA foci.

In a second approach, we tested whether expanded CCUG repeats would modify rbFOX1 splicing regulatory functions. Alternative splicing of the human mitochondrial ATP synthase gamma-subunit (*ATP5C1* also named F1gamma) exon 9 minigene is regulated by the rbFOX proteins[39]. However, overexpression of expanded CCUG repeats does not modify *ATP5C1* minigene splicing (Fig. 4c). As a control, overexpression of expanded CCUG repeats alters the splicing regulation of the insulin

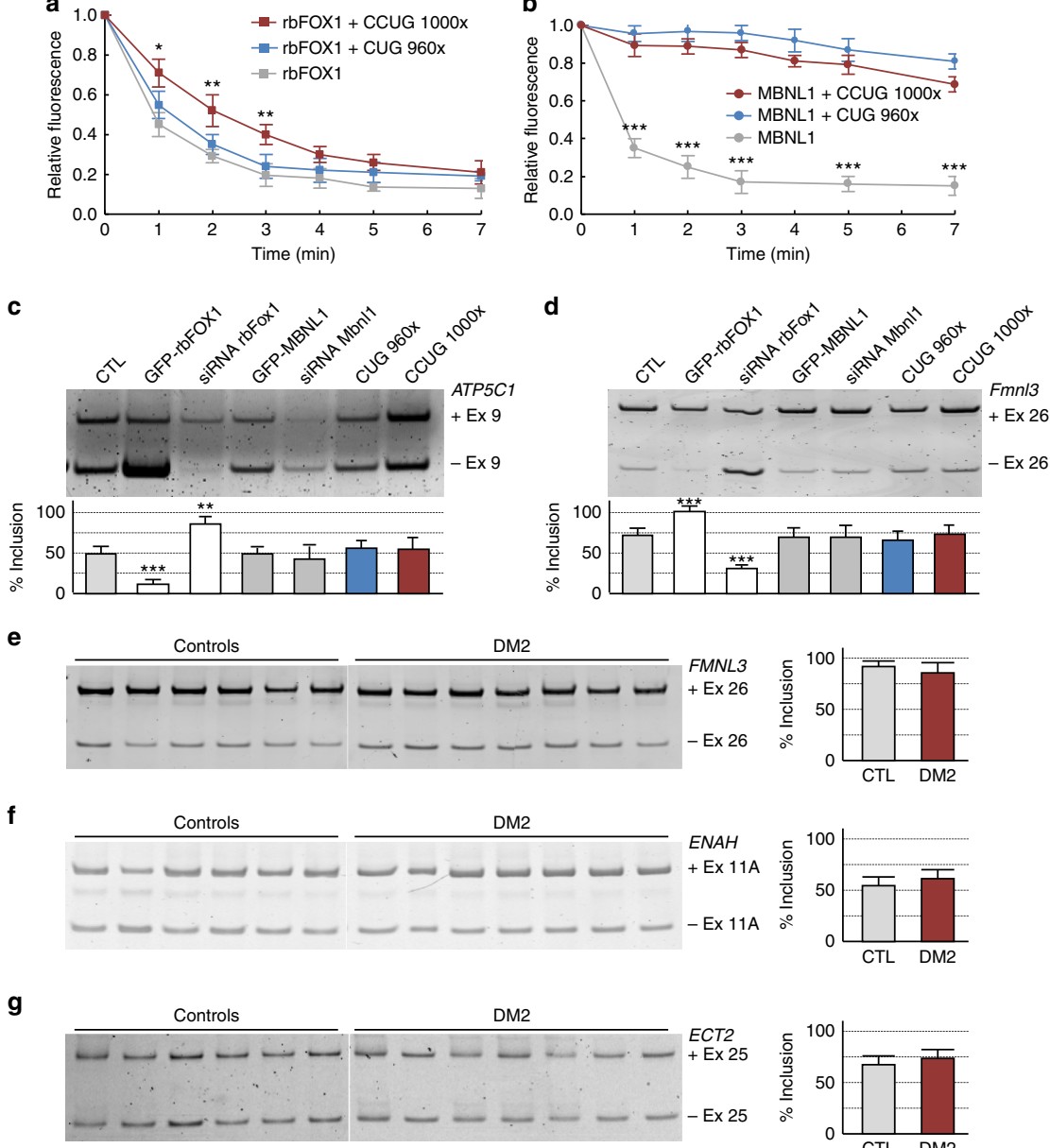

**Fig. 4** rbFOX1 is not sequestered within CCUG RNA foci. **a** Time course quantification of photoconverted spot of dendra2-rbFOX1 in COS7 cells co-transfected with a plasmid expressing dendra2-rbFOX1 and a plasmid expressing either no repeats (CTL), 960 CUG or 1000 CCUG repeats. Each data point is the average of 7 spot. **b** As in **a** but with dendra2-MBNL1. **c** Upper panel, RT-PCR analysis of RNA extracted from two days differentiated C2C12 cells co-transfected with a minigene expressing the exon 9 of the mitochondrial ATP synthase gamma-subunit gene and either with a plasmid expressing rbFOX1, MBNL1, 960 CUG repeats or 1000 CCUG repeats or with a siRNA directed against *rbFox1* or *Mbnl1*. Lower panel, quantification of exon 9 inclusion of transfected *ATP5C1* minigene. **d** Upper panel, RT-PCR analysis of endogenous *Fmnl3* exon 26 alternative splicing from GFP-FACS sorted C2C12 cells differentiated two days and co-transfected with a plasmid expressing eGFP and either with a plasmid expressing rbFOX1, MBNL1, 960 CUG repeats or 1000 CCUG repeats or with a siRNA directed against *rbFox1* or *Mbnl1*. Lower panel, quantification of *Fmnl3* exon 26 inclusion. **e–g** RT-PCR analysis (left panel) and quantification (right panel) of alternative splicing of *FMNL3*, *ENAH*, and *ECT2* performed on total RNA extracted from adult skeletal muscle of control or DM2 individuals. Error bars indicate s.e.m. of three independent experiments. Student's *t*-test, asterisk (*) indicates $p < 0.5$, asterisk (**) indicates $p < 0.01$, asterisk (***) indicates $p < 0.001$

receptor (*INSR*) exon 11 minigene (Supplementary Fig. 4A). Of interest, MBNL1 regulates *INSR* minigene splicing, but does not modulate *ATP5C1* minigene splicing (Fig. 4c and Supplementary Fig. 4A). These results suggest that expression of expanded CCUG repeats alters MBNL1 splicing regulatory function, but does not modify rbFOX1 splicing activity. Importantly, identical results were observed with endogenous alternative splicing events regulated by the rbFOX proteins[42]. Namely, overexpression of

expanded CCUG repeats has no significant effect on endogenous *Fmnl3* exon 26, *Tbx3* exon 25, and *Enah* exon 11 A alternative splicing (Fig. 4d and Supplementary Fig. 4B, 4C). We confirmed that overexpression or siRNA-mediated depletion of rbFox1 regulates *Fmnl3*, *Tbx3* and *Enah* alternative splicing, while expression or depletion of Mbnl1 has no effect (Fig. 4d and Supplementary Fig. 4B, 4C). Western blotting indicated correct siRNA-mediated depletion of rbFox1 and Mbnl1 proteins

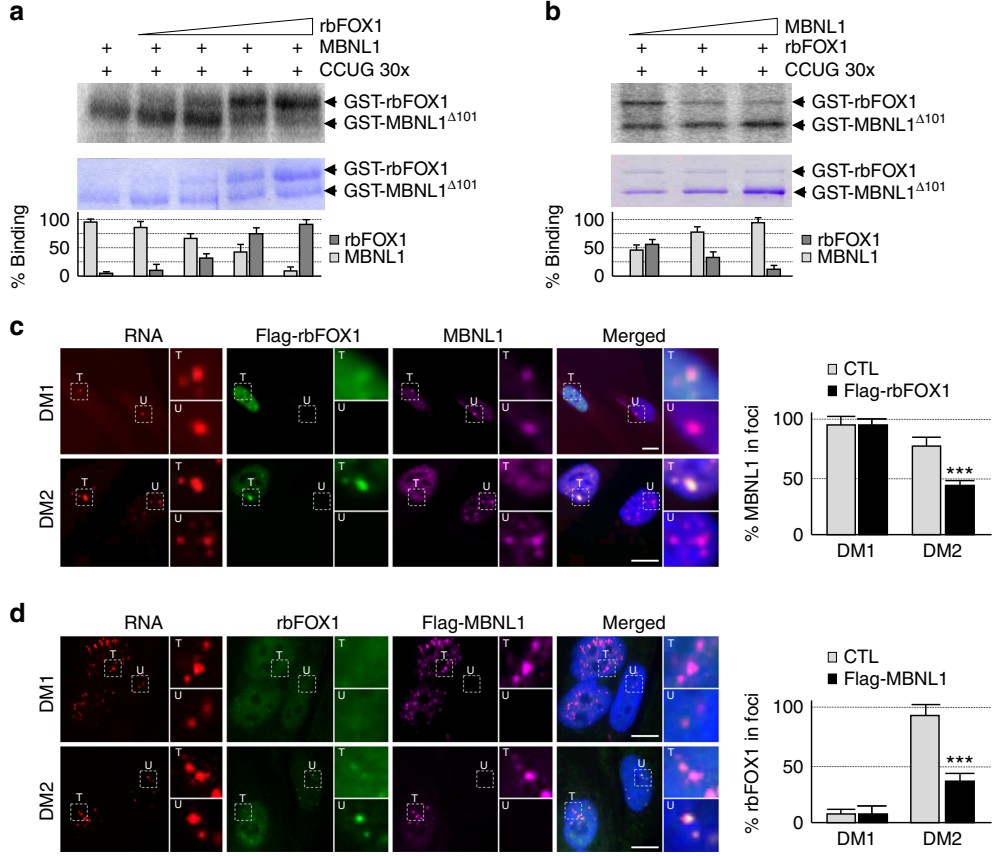

**Fig. 5** rbFOX1 competes with MBNL1 for binding to expanded CCUG repeats. **a** Upper panel, UV-cross-linking binding of 1 μg of GST-MBNL1$^{\Delta101}$ to 10,000 CPM of uniformly [αP$^{32}$] internally labeled in vitro transcribed RNA containing 30 CCUG repeats was competed by increasing amounts (0.25, 0.5, 1, and 2 μg) of GST-rbFOX1. Middle panel, loading of recombinant MBNL1 and of rbFOX1 proteins was verified by coomassie staining. Lower panel, quantification of the binding of MBNL1 and rbFOX1 to expanded CCUG repeats. **b** UV-cross-linking as in **a** but with binding of 0.5 μg of GST-rbFOX1 competed by 0.5, 1, and 2 μg of GST-MBNL1$^{\Delta101}$. **c** Left panel, RNA FISH of expanded CUG or CCUG repeats coupled to immunofluorescence against Flag-rbFOX1 and MBNL1 on primary cultures of myoblasts originating from muscle biopsies of individuals with either DM1 or DM2 and transfected with a plasmid expressing FLAG-tagged rbFOX1. Each image shows at least two different cells, one transfected (T) and one not transfected (U), with insets showing higher magnification focusing on RNA foci. Right panel, quantification of the signal of endogenous MBNL1 localized within the CCUG RNA foci. **d** As in **c** but with transfection of FLAG-tagged MBNL1 and immunofluorescence against rbFOX1 and Flag-MBNL1. Scale bars, 10 μm. Nuclei were counterstained with DAPI. Error bars indicate s.e.m. of three independent experiments. Student's *t*-test, asterisk (***) indicates *p* < 0.001

(Supplementary Fig. 4D). As a further control, expression of endogenous Mbnl1 is not altered upon overexpression or depletion of rbFOX1 (Supplementary Fig. 4E).

Finally, we tested whether rbFOX-dependent alternative splicing events were modified in skeletal muscle samples of DM2 patients. RT-PCR analysis demonstrated that exon 26 of *FMNL3*, exon 11 A of *ENAH*, and exon 25 of *ECT2* do not display any splicing abnormalities in muscle samples of individuals with DM2 compared to control skeletal muscle samples (Fig. 4e, g). As controls, alternative splicing of exons regulated by MBNL1, namely exon 29 of *CACNA1S* (CAV1.1), exon AS1 of *RYR1*, exon 22 of *ATP2A1* (SERCA), exon 7 of *MBNL1* and exon 11 of *LDB3* (CYPHER/ZASP), are all altered in DM2 muscle samples (Supplementary Fig. 4F, 4G, 4H, 4I, 4J). Of interest, the extend of splicing alterations in distal skeletal muscle samples is greater in DM1 compared to DM2 (Supplementary Fig. 4F, 4G, 4H, 4I, 4J), which correlates with the greater impairment of distal muscle in DM1 compared to DM2. These results indicate that, in contrast to MBNL1, the splicing regulatory functions of the rbFOX proteins are not altered by expression of expanded CCUG repeats.

**rbFOX compete the binding of MBNL1 to expanded CCUG repeats**. The binding of both MBNL1 and rbFOX1 to expanded CCUG repeats questions whether these interactions are mutually exclusive. In vitro, an excess of either rbFOX1 or rbFOX2 chases MBNL1 from binding to CCUG repeats (Fig. 5a and supplementary Fig. 5A). Conversely, an excess of MBNL1 competes rbFOX1 binding to CCUG repeats (Fig. 5b). In DM2 muscle cell cultures, overexpression of rbFOX1 reduces the quantity of MBNL1 present within RNA foci of expanded CCUG repeats and concomitantly increases the labeling of free diffuse MBNL1 (Fig. 5c). Inversely, shRNA-mediated depletion of rbFOX1 expression increases the localization of endogenous MBNL1 within CCUG RNA foci (Supplementary Fig. 5B). Consistent with a mutually exclusive competition mechanism, overexpression of MBNL1 partially displaces rbFOX1 from CCUG RNA foci in DM2 muscle cells (Fig. 5d). As controls, we observed no displacement of MBNL1 localization upon modulation of rbFOX1 expression in DM1 muscle cell cultures (Fig. 5c and Supplementary Fig. 5C).

These results suggest that rbFOX1 competes with MBNL1 to bind to CCUG RNA repeats. Nevertheless, it is possible that other

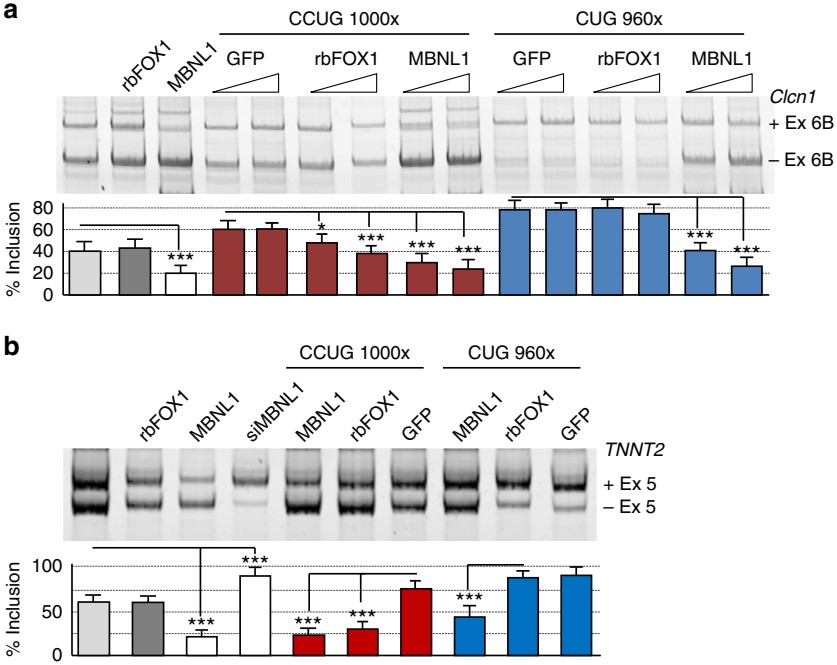

**Fig. 6** rbFOX1 corrects splicing alterations caused by CCUG repeats. **a** Upper panel, RT-PCR analysis of alternative splicing of the mouse chloride channel *Clcn1* exon 6B minigene co-transfected in C2C12 mouse muscle cells with a plasmid expressing either 960 CUG repeats or 1000 CCUG repeats and a vector expressing either rbFOX1 or MBNL1. Lower panel, quantification of *Clcn1* exon 6B inclusion. **b** As in **a** but with *TNNT2* (cTNT) exon 5 minigene. Error bars indicate s.e.m. of three independent experiments. Student's *t*-test, asterisk (*) indicates $p < 0.5$, asterisk (**) indicates $p < 0.01$, asterisk (***) indicates $p < 0.001$

mechanisms are at play. As noted previously[44,50,51], overexpression of rbFOX1 inhibits inclusion of *Mbnl1* exon 7 (Supplementary Fig. 5D). This 36 nts long exon is reported to modulate MBNL1 self-dimerization[52]. However, this splicing change is unlikely to contribute significantly to MBNL1 reduced localization within CCUG RNA foci upon rbFOX1 overexpression as in the same conditions of rbFOX1 overexpression we observed no changes in MBNL1 localization within CUG RNA foci in DM1 muscle cells (Fig. 5c and Supplementary Fig. 5C). Furthermore, rbFOX1 overexpression neither changes MBNL1 nuclear/ cytoplasmic localization (Fig. 5c), nor MBNL1 global expression (Supplementary Fig. 5E). Similarly, expressions of *rbFOX1*, *rbFOX2*, *MBNL1*, and *MBNL2* mRNAs are not altered in muscle samples of individuals with DM2 compared to non-DM controls or individuals with DM1 (Supplementary Fig. 5F).

A potential mechanism of competition requires that a sufficient quantity of endogenous rbFOX1 is expressed to prevail over MBNL1. In human skeletal muscle, *MBNL1* mRNA is expressed at slightly higher level compared to *rbFOX1* mRNA with 50 to 70 Transcripts Per Kilobase Million (TPM) for *MBNL1* compared to 30 to 40 TPM for *rbFOX1*, while *MBNL2* and *rbFOX2* mRNAs are expressed at similar levels (20 to 30 TPM)[35,36]. However, western blotting analysis indicated that both rbFOX1 and MBNL1 are expressed at similar levels in control or in DM2 adult human skeletal muscle samples (Supplementary Fig. 5G). To avoid a bias due to antibody differences, recombinant purified proteins were used as standards (Supplementary Fig. 5H). Thus, a competition between MBNL1 and rbFOX1 is possible as these proteins are expressed at comparable levels in human adult skeletal muscle.

**Expression of rbFOX1 corrects splicing alterations of CCUG repeats.** Titration of the MBNL proteins within CUG or CCUG RNA foci leads to specific alternative splicing changes. Hence, we tested whether the release of MBNL1 from CCUG RNA foci upon

rbFOX1 overexpression may correct splicing alterations typical of myotonic dystrophy. Importantly, overexpression of rbFOX1 in C2C12 muscle cells partly corrects splicing alterations of the chloride channel (*Clcn1*) and of the cardiac troponin T (*TNNT2*) minigenes induced by overexpression of expanded CCUG repeats (Figs. 6a, b and Supplementary Fig. 6A). Inversely, siRNA-mediated decreased expression of endogenous rbFox1 increases the pathogenic effect of expanded CCUG repeats on *Clcn1* and *TNNT2* minigene splicing (Supplementary Fig. 6B, 6C). In contrast, overexpression or decreased expression of rbFOX1 has no correcting effect on *Clcn1* and *TNNT2* splicing changes induced by expanded CUG repeats (Fig. 6a, b and Supplementary Fig. 6B and 6C). As a positive control, expression of MBNL1 corrects splicing alterations caused by either CUG or CCUG expanded repeats (Fig. 6a, b). Note that MBNL1 regulates alternative splicing of the *Clcn1* and *TNNT2* minigenes, while rbFOX1 has no effect in absence of CCUG repeats (Fig. 6a, b). In support of a splicing regulation directly modulated by MBNL1 but not by rbFOX1, gel-shift experiments indicate that recombinant purified MBNL1, but not rbFOX1, binds to *Clcn1* and *TNNT2* minigenes (Supplementary Fig. 6D). As a further control, western blotting indicates that overexpression of rbFOX1 does not modify endogenous Mbnl1 levels (Supplementary Fig. 6E).

**Expression of rbFOX1 alleviates muscle atrophy.** To test whether expression of rbFOX1 could rescue any deleterious phenotypes caused by expression of expanded CCUG repeats, we developed *Drosophila* models of DM1 and DM2 that overexpress rbFOX1. Expression of uninterrupted expanded CTG or CCTG repeats deleted of their natural *DMPK* or *CNBP* sequences were targeted to fly muscle through crossing *UAS*-CTG or *UAS*-CCTG flies with a *Myosin heavy chain* (*Mhc*)-*GAL4* driver line. As noted previously[37], expression of expanded CUG or CCUG repeats leads to muscle dysfunctions compared to a control GFP line or to *Drosophila* lines carrying control (20×) CTG or CCTG repeats

(Supplementary Fig. 7A). We then generated recombinant *Drosophila* flies expressing GFP-tagged rbFOX1 under the *UAS* promoter and crossed these flies with either DM1 or DM2 lines. Importantly, overexpression of rbFOX1 fully rescues muscles atrophy caused by expression of expanded CCUG repeats (Fig. 7a). Two different independent *Drosophila* lines expressing rbFOX1 were tested and gave comparable results (Fig. 7a). Similarly, a second independent line expressing expanded CCUG repeats was tested and shows identical muscle correction upon overexpression of rbFOX1 (Supplementary Fig. 7B). Next, we

tested the effect of rbFOX1 in DM1 flies. Interestingly, overexpression of rbFOX1 does not suppress muscle atrophy in *Drosophila* expressing expanded CUG repeats (Fig. 7b). As a positive control, expression of MBNL1 fully corrects muscle alterations in DM1 and DM2 flies (Figs. 7a, b). Consistent with correction of muscle atrophy, functional assays demonstrated that overexpression of rbFOX1 rescues both flying (Fig. 7c) and climbing (Fig. 7d) defects caused by expression of expanded CCUG repeats. This correction is specific, as overexpression of rbFOX1 has no rescue effect in CUG expressing flies (Figs. 7c, d).

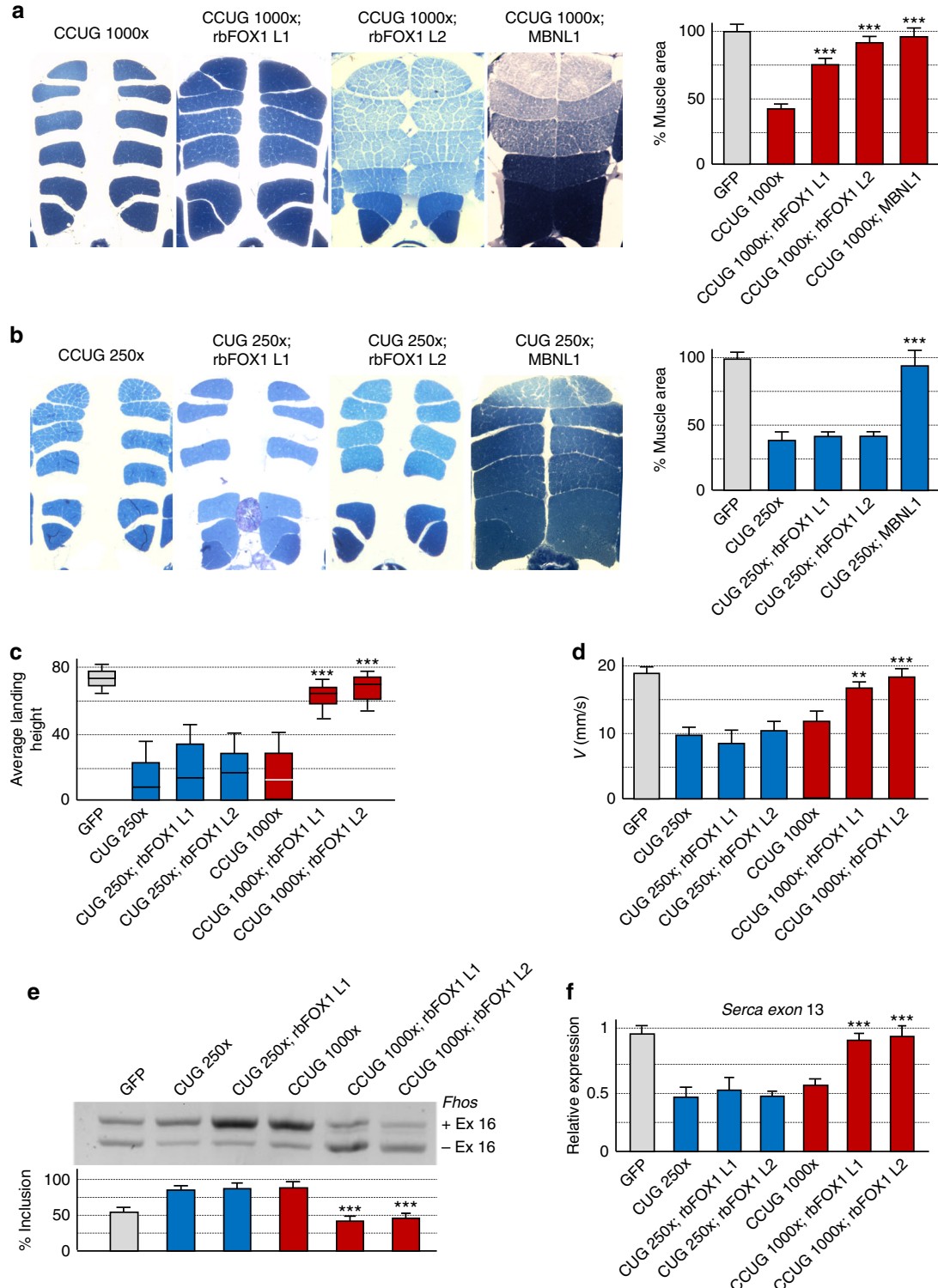

**Fig. 7** rbFOX1 alleviates the symptoms caused by expanded CCUG repeats. **a** Left panel, representative dorsoventral sections of resin-embedded adult thoraces showing indirect flight muscles (IFMs) of flies expressing expanded CCUG repeats and GFP-tagged rbFOX1 or MBNL1. Right panel, IFM muscle area quantification. **b** As in **a** but with DM1 flies expressing expanded CUG repeats. **c** Notched box plot showing the median and the distribution of average landing height (cm) data from flight assay of control, expanded CUG and expanded CCUG expressing flies with or without expression of GFP-rbFOX1. The horizontal lines within the boxes represent median values, whereas bottom and top edges of the boxes represent the 25th and 75th percentiles and bottom and top whiskers reach the 10th and 90th percentiles, respectively. **d** Quantification of climbing speed in mm/s of control, expanded CUG and expanded CCUG expressing flies with or without expression of GFP-rbFOX1. **e** Upper panel, RT-PCR analysis of endogenous *Fhos* exon 16 in flies expressing expanded CUG or CCUG repeats and GFP-rbFOX1. Lower panel, quantification of *Fhos* exon 16 inclusion. **f** Quantification of *Serca* exon 13 expression in flies expressing expanded CUG or CCUG repeats and GFP-rbFOX1. Detection of endogenous *Rp49* gene expression was used for normalization. Expression of transgenes in muscle was driven by Mhc-GAL4. Error bars indicate means ± s.e.m. of three to five independent experiments. Student's *t*-test, asterisk (**) indicates $p < 0.01$, asterisk (***) indicates $p < 0.001$

At the molecular level, titration of *Drosophila* Muscleblind (Mbl) within CUG or CCUG RNA foci leads to specific mRNA splicing alterations[37]. Importantly, overexpression of rbFOX1 corrects *Fhos* and *Serca* splicing alterations caused by expression of expanded CCUG repeats, but has no correcting effect in DM1 flies (Fig. 7e, f). As controls, *Fhos* and *Serca* alternative splicing were not altered in GFP-tagged rbFOX1 flies compared to control GFP transgenic *Drosophila* (Supplementary Fig. 7C, 7D). Furthermore, RNA FISH coupled to immunofluorescence indicated that rbFOX1 co-localizes with RNA foci of expanded CCUG repeats, but not with RNA foci of expanded CUG repeats (Supplementary Fig. 7E). Concomitant labeling of endogenous Mbl showed that Mbl co-localizes with RNA foci of both expanded CUG and CCUG repeats (Supplementary Fig. 7E).

Finally, we noted that the longevity of *Drosophila* expressing either expanded CUG or CCUG repeats was reduced compared to control or GFP-expressing flies. Strikingly, overexpression of rbFOX1 partly rescues the decreased lifespan of DM2 flies, but has no effect in DM1 flies (Fig. 8a, b). Overall, these results indicate that rbFOX1 can correct splicing alterations, muscle atrophy, locomotor defects, as well as reduced lifespan in an animal model of DM2.

## Discussion

An RNA gain-of-function mechanism for myotonic dystrophy predicts that an increased quantity or a higher number of expanded CUG or CCUG repeats should lead to a greater titration of the MBNL RNA binding proteins, ultimately resulting in increased severity of the disease. This model is consistent with the increased number of CTG repeats, increased splicing alterations and increased disease severity observed in congenital CDM1 compared to the adult-onset DM1 form[23]. However, this RNA gain-of-function model is challenged by the milder severity of DM2 compared to DM1, despite a higher number of expanded repeats in DM2 compared to DM1. This paradox suggests the existence of both common and distinct mechanisms involved in the pathogenesis of DM1 and DM2. The most likely explanation is that expanded CCUG repeats are inherently less pathogenic due to their genomic localization. Indeed, expanded CCUG repeats are located within the first intron of the *CNBP* gene, which presumably results in a lesser RNA stability and ultimately a lesser expression of the CCUG repeats compared to the CUG repeats embedded within the 3'UTR of the *DMPK* mRNA. However, RNA FISH analyses revealed that CCUG RNA foci are 8 to 13 folds more intense than CUG RNA foci in skeletal muscle tissues of individuals with myotonic dystrophy[7], suggesting that RNA foci are present at pathogenic levels in DM2. A second mechanism possibly contributing to the increased toxicity of the CUG repeats is the higher phosphorylation and expression of CUGBP1 reported in DM1 hearts, especially in the most severely affected individuals[5]. However, a higher expression of CUGBP1 is also reported in DM2 samples[53,54], although inconsistently[33,55].

Thirdly, repeat-associated non-ATG (RAN) translation of CAG repeats from antisense *DMPK* transcripts results in expression of potentially toxic proteins in type 1 myotonic dystrophy[56]. However, RAN translation of sense CCUG and antisense CAGG expanded repeats in brain samples of individuals with DM2 brain is also reported[57]. It remains to be determined whether RAN translation is deleterious or plays a protective role in respect to MBNL1 titration in myotonic dystrophy[56,57]. Finally, a difference of toxicity between expanded CUG and CCUG repeats based on a difference in MBNL1 affinity is unlikely as MBNL1 binds in vitro to CCUG repeats with a slightly better affinity compared to CUG repeats[58–60].

In the present work, we found that rbFOX1 binds to CCUG repeats and that its overexpression partly reduces MBNL1 titration, splicing alterations and phenotype severity in cell and animal models of DM2. These data suggest a model where at the time of CCUG repeats transcription, rbFOX proteins compete with MBNL proteins for the binding to the nascent CCUG RNA transcript, thus partially reducing MBNL1 titration (Fig. 8c, d). A reduced titration of MBNL1 by expanded CCUG repeats is consistent with the lesser splicing and differentiation defects reported in DM2 muscle cell cultures compared to DM1 cells[32–34], and with the lesser alternative splicing changes that we observed in DM2 distal muscles compared to DM1 distal muscle samples. These data are also consistent with the absence of expression of CG32062, the *Drosophila* ortholog of the rbFOX proteins, in adult fly skeletal muscles, which may explain the absence of protection against expanded CCUG repeats in DM2 fly models. In conclusion, our data reinforce a model of RNA gain-of-function for myotonic dystrophy as a lesser toxicity of the CCUG expanded repeats can be partly explained by a lesser titration of the causative MBNL RNA binding proteins.

This work raises several questions. First, rbFOX proteins regulate the alternative splicing of *MBNL1* pre-mRNA[44,50,51]. Indeed, overexpression of rbFOX1 in our cell systems changes alternative splicing of *MBNL1* exon 7, a 36 nts exon contributing to MBNL1 dimerization[52]. In contrast, rbFOX1 does not alter alternative splicing of *Mbnl1* exons 3 and 5, which are 204 and 54 nts long exons regulating MBNL1 binding to RNA and nuclear localization, respectively. Of interest, we observed no overt effect of rbFOX1 overexpression on the global expression or the nuclear/ cytoplasmic localization of MBNL1. Furthermore, overexpression of rbFOX1 has a correcting splicing effect only in CCUG expressing cells and not in control or in CUG expressing cells. Thus, while we cannot formally exclude an indirect contribution of MBNL1 exon 7 splicing change or other mechanisms upon rbFOX1 overexpression, these results favor a model where rbFOX1 overexpression partly competes MBNL1 away from binding to CCUG repeats (Figs. 8c, d).

A second intriguing point is that rbFOX proteins are well known to bind to the UGCAUGY RNA sequence. Here, we found that rbFOX1 can also recognize CCUG expanded repeats.

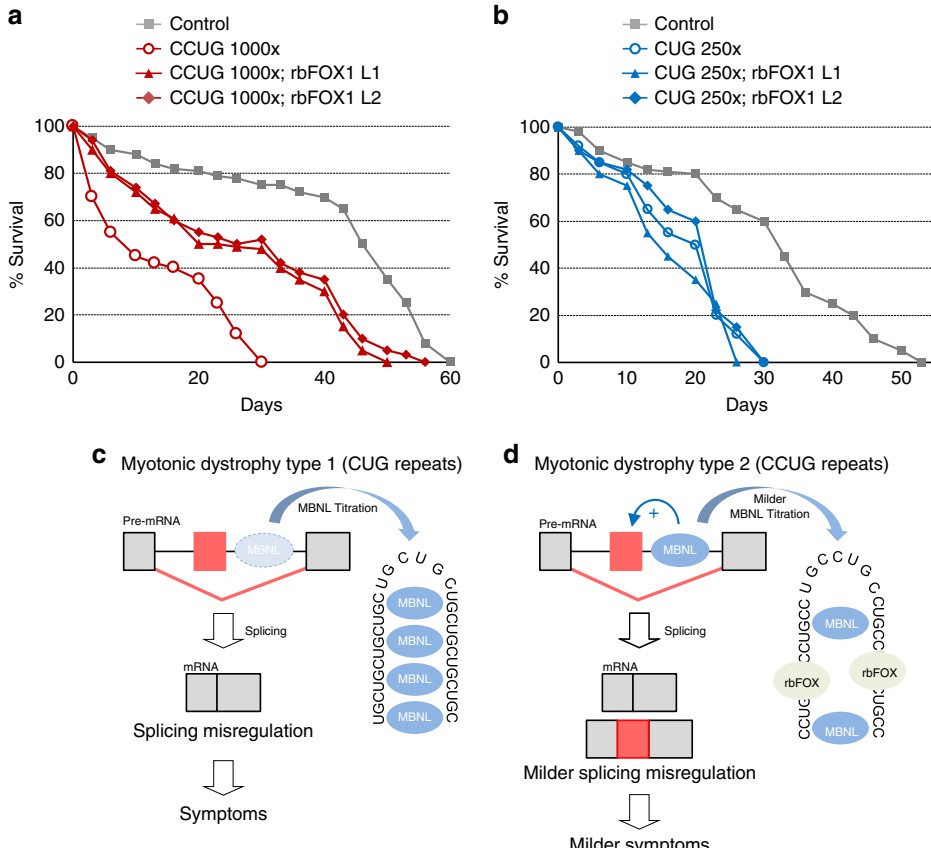

**Fig. 8** Model of splicing alteration in myotonic dystrophy. **a**, **b** Lifespan in days of control and expanded CCUG (**a**) or CUG (**b**) expressing flies with or without expression of GFP-tagged rbFOX1. Flies expressing expanded CCUG or CUG repeats showed a median life of 10 and 21 days, respectively, which was significantly reduced in comparison to control flies ($n = 50$ flies, $p < 0.0001$, Gehan-Breslow-Wilcoxon test). Expression of rbFOX1 partly corrects median life and lifespan in CCUG expressing flies ($n = 50$ flies, CCUG rbFOX1 L1 $p < 0.0001$ and CCUG rbFOX1 L2 $p = 0.0002$). In contrast, expression of rbFOX1 does not improve lifespan of CUG expressing flies. **c** In myotonic dystrophy type 1, expanded CUG repeats represent an excess of ligand for MBNL proteins, resulting in their titration and sequestration in discrete nuclear RNA foci, ultimately resulting in alternative splicing changes that are inadequate to adult physiology. **d** In myotonic dystrophy type 2, expanded CCUG repeats also titrate MBNL proteins, but with a lesser efficiency due to competition with rbFOX proteins, which ultimately results in milder alternative splicing changes and milder symptoms compared to DM1

However, rbFOX1 binding to a single UGCCUGC motif was barely detectable compared to binding to the UGCAUGC RNA motif. Thus, even if the structure of rbFOX1 can accommodate a central cytosine instead of an adenosine, the binding of rbFOX1 to a single UGCCUGC motif is weak and likely not sufficient to regulate alternative splicing. In contrast, the extensive repetition of that RNA motif in DM2 patients, in which up to 11,000 CCUG repeats can be expressed, may compensate for the low affinity of rbFOX1 toward a unique UGCCUGC RNA motif. As a further cautionary note, we tested direct binding of rbFOX1 to CCUG repeats only in vitro and on expansions of limited size (10, 30 and 100 repeats). Thus, direct binding of rbFOX1 to large CCUG expansions in individuals with DM2 will require a formal demonstration, such as rbFOX1 CLIP experiments on human DM2 tissue samples.

A third puzzling result is the absence of sequestration of rbFOX1 within CCUG RNA foci. Indeed, splicing events regulated by the rbFOX proteins are neither altered in CCUG-expressing cells nor in skeletal muscle samples of individuals with DM2. Similarly, photoconversion analysis indicated that rbFOX proteins are not immobilized within RNA foci of expanded CCUG repeats. These results are consistent with previous reports demonstrating that a RNA binding protein can be enriched in nuclear RNA foci without being immobilized within such structure[61,62]. These data confirm that an image of co-localization is

not an evidence of immobilization and that functional sequestration should be experimentally tested. The mechanisms underlying "liberty" of rbFOX1 but "captivity" of MBNL1 within CCUG RNA foci are unclear. A possible explanation is that the relative affinity of rbFOX1 for its endogenous pre-mRNA targets, which contain UGCAUGY RNA motifs, is higher compared to its affinity for expanded CCUG repeats. In contrast, MBNL1 presents a higher affinity for CUG or CCUG repeats compared to its normal pre-mRNA targets. Moreover, it is also possible that the different RNA binding architectures between rbFOX and MBNL proteins may play a role in modulating their sequestration, as MBNL1 contains four zinc fingers binding to four YGC RNA motifs, while rbFOX1 binds to one UGCAUGY sequence through its single RRM domain.

An additional argument against a functional titration of rbFOX proteins by expanded CCUG repeats in DM2 is the study on animal models and individuals with loss-of-function mutations of rbFOX1 or rbFOX2. In mouse models, knockout for *rbFox1* or *rbFox2* present severe skeletal muscle, heart and brain dysfunctions[51,63–66], which are different from the alterations observed in myotonic dystrophy. In contrast, transgenic mice expressing expanded CUG repeats present similar splicing alterations and phenotypes than *Mbnl1* and/ or *Mbnl2* knockout mice[10,13,14,20,23,67]. In humans, rbFOX1 binds to mRNAs that are enriched in pathways involved in cortical neuronal

development[45,68], and mutations or deletions within the *rbFOX1* gene have been identified in patients with neurological disorders including epilepsy, schizophrenia, mental retardation and autism[69–71], which are different from the symptoms observed in DM2. Similarly, loss-of-function mutations in the *rbFOX2* gene are responsible for developmental cardiac alterations[72–74], which are different from the adult-onset progressive cardiac conduction defects and arrhythmias observed in individuals with myotonic dystrophy. Thus, human genetics and animal models argue against a loss of rbFOX functions in myotonic dystrophy type 2.

Lastly, our study was restricted to skeletal muscle cells and tissue samples. Hence, it remains to be determined the influence of rbFOX proteins on the titration of MBNL1 and MBNL2 in heart, as well as in neurons where rbFOX3 (NeuN) is highly expressed. Similarly, whether the competition between rbFOX and MBNL proteins may contribute to the absence of a congenital form in DM2 remains to be tested. We also want to emphasize that our study does not preclude that other mechanisms participate to the lesser severity of DM2.

In conclusion, these results confirm that the sequestration of MBNL1 within CUG or CCUG RNA foci is a binding titration that can be reversed through competition with antisense oligonucleotides[75], small molecules[76,77], or RNA binding proteins[24]. These various approaches may hopefully open some route for therapeutic approaches in myotonic dystrophy.

## Methods

***Drosophila* genetics and functional analysis**. *Mhc-GAL4* and *UAS-MBNL1* lines were previously described[78]. *UAS-(CTG)20x*, *UAS-(CTG)250x*, *UAS-(CCTG)20x* and *UAS-(CCTG)1000x* transgenic lines were previously described[37]. The line *UAS-rbFOX1-GFP* was generated by injecting the plasmid into *w1118* embryos by BestGene Inc. For simultaneous expression of rbFOX1 or MBNL1 and CTG or CCTG expanded repeats, we generated recombinant flies *UAS-rbFOX1 UAS-repeats* and *UAS-MBNL1 UAS-repeats*. We expressed UAS-GFP transgene simultaneously with the expanded repeats as a control of transgene dosage for comparisons in the rescue experiments. Expression of the transgenes in muscle was achieved by crossing the recombinants flies to *Mhc-GAL4*. All flies were maintained at 25 °C with standard food. Longevity assays, flying assays, RNA extraction, RT-PCR and qRT-PCR, as well as processing of *Drosophila* tissues for fluorescent and histological assessments were performed as described[37]. RNA foci detection was performed using either CAG8x- or CAGG8x-Cy3 DNA oligonucleotide probes (Sigma). Rabbit anti-GFP (Roche) and sheep-anti-Mbl antibodies were used to detect rbFOX1-GFP and endogenous Mbl expression in fly muscle, respectively. Groups of ten 5-day-old males were transferred into vials of 1.5 cm in diameter and 25 cm in height and were tested for climbing velocity after a period of 24 h without anesthesia. The height reached from the bottom of the vial by each fly in a period of 10 s was recorded with a camera. For each genotype, approximately 30 flies were tested. The results show the mean speed in mm/s.

**Plasmids and constructions**. Interrupted 960 CTG repeats, deprived of *DMPK* sequence, were subcloned from the DT960 plasmid[9] into pCDNA3.1. Pure CTG and CCTG expanded repeats were generated by PCR amplification of self-priming single-stranded CTG and CAG or CCTG and CAGG oligonucleotides as described[79]. Synthesized DNA duplexes were electrophoresed, size fractionated, purified by DNA gel extraction kit (Qiagen), 5'-phosphorylated with T4 polynucleotide kinase and cloned into the *Eco*RV site of pcDNA3.1 or pUAS plasmids. The recombinant plasmids containing interrupted stretches of 250 CTG or 1,000 CCTG repeats were amplified in STBL3 (Invitrogen) *E. coli* strain at 20 °C. Plasmid DNA was purified using Qiagen plasmid DNA purification kit and sequenced from either end to ensure sequence integrity of the clones. MBNL1 and rbFOX1 cDNAs are encoding full-length muscle isoform proteins with their respective Nuclear Localization Signal (NLS), corresponding to MBNL1 42 kDa isoform (Accession # NM_207293.1; transcript variant #3) that includes the exon linker between the four zinc fingers and the exon 6 of 54 nts encoding part of MBNL1 NLS, and the rbFOX1 transcript variant #2 (Accession # NM_145892.2) encoding the full RRM, the muscle M43 exon and its FAPY end involved in nuclear localization. MBNL1 and rbFOX1 cDNAs were cloned in frame downstream of the GST tag in pET28 vector, downstream of either the eGFP-tag or of the Flag-tag in classic eukaryotic or *Drosophila* expression vectors. Concerning recombinant proteins, truncated cDNA of MBNL1 (MBNL1$^{\Delta101}$ containing residues 1 to 253) contains the full N-terminal RNA binding domain of MBNL1 but lacks its C-terminal part that is insoluble and impairs MBNL1 expression and purification in bacteria. CTG (#63087), CCTG (#63088), rbFOX1 (#63085) and MBNL1 (#61277) expression plasmids have been deposited on Addgene.

**Patient and cell samples**. Muscle biopsies and primary human myoblast cells originating from muscle needle biopsies of six non-DM control individuals, seven individuals with DM1 and seven individuals with DM2 were described previously[18,27,30,32–34,55]. All skeletal muscle biopsies, taken either from the anterior tibial or gastrocnemius muscles were performed for diagnostic purpose and were obtained from patients genetically confirmed heterozygous for the DM1 or DM2 expansion, as well as from control individuals. All studies were done with the approval of the Institutional Review Board of the Ludwig-Maximilians University in Munich, of the Neuromuscular Research Center of the Tampere University Hospital, of the IRCCS Policlinico San Donato and of the Toneyama National Hospital. Informed consent was obtained from all human participants.

**nanoLC-MS/MS analysis**. One milligram of nuclear proteins extracted from mouse brain or C2C12 muscle cells differentiated four days were passed over synthetic biotinylated CCUG30x or CUG30x RNA (30 nmoles, Thermo Scientific) bound to streptavidin coated magnetic beads (Dynabeads M-280 streptavidin, Invitrogen) in the presence of 20 mM Hepes, 300 mM NaCl, 2 mM MgCl2, 0.01% NP40, 1 mM DTT and protease inhibitor (PIC, Roche). The magnetic beads with immobilized RNA and their bound proteins were washed three times with the binding buffer and bound proteins were eluted by boiling 3 min in sample buffer prior to 4–12% SDS-PAGE (NuPAGE 4–12% bis-Tris Gel, Invitrogen) separation and silver staining (SilverQuest, Invitrogen). Protein bands were excised from the gel, digested and identified using NanoESI_Ion Trap (LTQ XL Thermo Fisher Scientific).

**Recombinant protein production and purification**. E. coli BL21(RIL) pRARE competent cells (Invitrogen) were transformed with pet28a-GST-rbFOX1, pet28a-GST-rbFOX2 or pet28a-GST-MBNL1$^{\Delta101}$, grown at 30 °C in 400 ml of LB medium supplemented with Kanamycin until an OD600 of 0.5, then 0.5 mM IPTG was added and the culture was further incubated 4 h at 30 °C. Harvested cells were sonicated in 50 mM Tris-Cl pH 7.5, 300 mM NaCl, 5% glycerol, 1 mM DTT, 5 mM EDTA, centrifuged 20 min at 20000 g and recombinant GST-tagged proteins were purified using the GST-Bind$^{TM}$ Kit (Novagen).

**In vitro RNA transcription**. Plasmids were linearized by *Eco*R1 restriction and 100 ng were transcribed using T7 transcription kit (Ambion) in presence of 1 µl of [$\alpha$P$^{32}$]-UTP (Perkin Elmer), analyzed on 8% denaturing polyacrylamide and quantified with LS-6500 counter (Beckman). Non-labeled transcripts were synthesized using the Megascript T7 kit (Ambion). After transcription, 1 unit of DNase I (Invitrogen, Carlsbad, CA) was added, and the sample was incubated for additional 30 min at 37 °C. Transcribed RNAs were then purified by micro Bio-Spin 6 chromatography columns (Bio-rad) according to the manufacturer's instructions. The sizes of RNAs were checked by gel electrophoresis on a denaturing 6% polyacrylamide gel.

**Gel shift assays**. 10 pM (3000 CPM) of labeled RNA was incubated at 90 °C for 5 min in binding buffer (BB, 0.75 mM MgCl2, 50 mM Tris-HCl (pH 7.0), 75 mM NaCl, 37.5 mM KCl, 5.25 mM DTT, 0.1 mg/mL BSA, 0.1 mg/mL Bulk tRNA) and allowed to cool to room temperature. After cooling, RNAsin was added to a final concentration of 0.4 U/µl. GST-MBNL1$^{\Delta101}$, GST-rbFOX1 or GST-rbFOX2 were then added and the mixture was incubated on ice for 20 min. The solution mixture was loaded onto a non-denaturing 6.0% (w/v) polyacrylamide gel (acrylamide/ bisacrylamide, 40:1, w/w) containing 0.5 × TBE (1 × TBE is 90 mM Tris-base, 89 mM Boric acid and 2 mM EDTA (pH 8.0)), which had been pre-electrophoresed at 110 V for 20 min. at 4 °C. The gel was electrophoresed at 110 V at 4 °C for 3 h, then dried and exposed to a phosphorimager screen and imaged using a Typhoon 9410. The data were fit to the following equation: $y = \min + ((\max-\min)/(1 + (x/\text{IC50})\text{-HillSlope}))$ where $y$ is the percentage of RNA bound, $x$ is the concentration of protein, min and max are the minimum and maximum percentage of RNA bound to MBNL1 or rbFOX1 or rbFOX2 (0-100%) and IC50 is the concentration where 50% of maximum binding is achieved.

**UV-crosslinking assays**. Twenty microgram of nuclear extract from C2C12 muscle cells differentiated four days or 0.5 to 1 µg of recombinant GST-MBNL1$^{\Delta101}$, GST-rbFOX1 or GST-rbFOX2 protein were incubated with 100 pM (30,000 CPM) of in vitro transcribed RNAs containing either 30 CUG or 30 CCUG repeats of uniformly [$\alpha$P$^{32}$] internally labeled, in 15 µl of 100 mM NaCl, 20 mM Hepes, 8 mM MgCl2, 0.1 mg/ml BSA, 0.1 mg/ml heparin and 0.5 mM DTT during 15 min at 30 °C. Reactions were transferred on parafilm and irradiated 5 min, on ice, at 2.5 cm of a UV lamp (Vilberloumat VL-100C). 1 µg of RNase A1 was added and incubated 30 min at 37 °C. RNA-protein complexes were resolved on 10% SDS-PAGE, coomassie labeled and analyzed using BAS-MS imaging plate (Fuji) and Typhoon 8600 imager (Molecular Dynamics).

**NMR and in silico energy minimization**. 1H-15N HSQC spectra were recorded using recombinant expressed 15N-labeled rbFox1-RRM (residues 108–216) on a Bruker Avance III 700 MHz spectrometer. All experiments were performed at 310 K in buffer containing 20 mM NaCl and 20 mM NaH2PO4 at pH 6.5 and 5% D2O.

Spectra were processed in Topspin 3.0 and visualized using Sparky 3.114. Initial coordinates were obtained by in silico mutation in PyMol using rbFox1-RRM bound to UGCAUGU (pdb code 2ERR) established previously[49]. Subsequent energy minimization was performed in AMBER 12. An unrestraint minimization was applied using the conjugate gradient method (XMIN) with a 0.01 convergence criterion for the root-mean-square.

**Cell cultures, transfections and transduction.** C2C12 cells were grown in DMEM, 1 g/1 glucose, 20% fetal calf serum and gentamycin at 37 °C, 5% (v/v) CO2. C2C12 were transfected with Lipofectamine 2000 (Invitrogen) according to the manufacturer's instructions with control plasmid or plasmid expressing 1,000 CCUG repeats. Six hours after transfection, the medium was removed and cells were differentiated two days in DMEM, 1 g/1 glucose, 2% fetal calf serum, insulin 10 µg/ml and gentamycin medium before RNA-FISH coupled to immuno-fluorescence. For endogenous splicing assays, C2C12 cells were co-transfected using Lipofectamine 2000 (Invitrogen) with a plasmid expressing GFP and a plasmid expressing either rbFOX1, MBNL1, 960 CUG repeats or 1,000 CCUG repeats. GFP positive cells were isolated two days after differentiation by using FACS Diva. For minigene splicing assays, C2C12 cells were co-transfected using Lipofectamine 2000 (Invitrogen) with a plasmid expressing the INSR or F1gama minigene and a plasmid expressing rbFOX1, MBNL1, 960 CUG repeats or 1000 CCUG repeats and differentiated 2 days before analysis. siRNA against rbFox1 or Mbnl1 were transfected by using RNAimax (Invitrogen) according to manufacturer's instruction. Primary human myoblast cells originate from muscle biopsies of genetically confirmed DM1 and DM2 patients. Myoblast cells were maintained at 37 °C with 10% CO2 in skeletal muscle cell basal media with supplements (PromoCell, Heidelberg, Germany) and 10% fetal calf serum. For myoblast differentiation, cells were maintained in DMEM with 2% fetal calf serum. For competition experiments, primary culture of muscle cells from DM2 patient was grown in DMEM, 1 g/1 glucose, medium 199 (4:1), 20% fetal calf serum and gentamycin at 37 °C, 5% (v/v) CO2 and transfected with a control pCDNA3.1 plasmid or a plasmid expressing FLAG tagged rbFOX1 according to Lipofectamine 2000 manufacturer's instructions. Cells were differentiated in DMEM, 1 g/1 glucose, medium 199 (4:1), 2% fetal calf serum, insulin 10 µg/ml and gentamycin five days before analyze. Transductions at a MOI of 1000 with recombinant adenovirus expressing a shRNA either directed against the luciferase or against rbFOX1 or MBNL1 were done on primary culture of muscle cells from DM patient differentiated two days. Medium was replaced 24 h after infection, and cells were analyzed 48 h after infection.

**Photoconversion analysis.** Live imaging was performed on a Nikon TiE inverted microscope (Nikon Instruments, Japan) equipped with a Tokai incubation chamber (Tokai Hit, Japan). We used a Nikon 100 × Apo TIRF NA 1,49 objective, image acquisition was done with a Yokogawa CSU-X1 spinning disk (Yokogawa Electric Corporation, Japan) coupled to a Photometrics Evolve 512 camera (Photometrics, Tucson, USA). A foci of Dendra2 tagged protein was photoconverted at 405 nm with a Roper Sientific ILas2 FRAP unit using Frap on Fly mode. In absence of foci, namely MBNL1 or rBFOX1 in control conditions or rbFOX1 with expanded CUG repeats, a 1 µm diameter spot of Dendra2-tagged protein was randomly chosen within the nucleoplasm and photoconverted. Dendra2-MBNL1 or Dendra2-rbFOX1 was imaged every thirty seconds after photoconversion with a 561 nm laser. The system was controlled by Metamorph software. For image analysis, mean intensity of the aggregate was measured over time using Fiji, and background and photobleaching corrections were applied.

**RNA FISH combined with immunofluorescence.** Frozen human muscle sections were incubated in PBS 1× during 10 min before RNA FISH. Glass coverslips containing plated cells or human sections treated as described above were fixed in PFA 4% during 15 min and washed two times with PBS. The coverslips or slides were incubated for 10 min in PBS plus 0.5% Triton X-100 and washed three times with PBS before pre-hybridization in 40% DMSO, 40% formamide, 10% BSA (10 mg/ml), 2× SCC for 30 min. The coverslips or slides were hybridized for 2 h in 40% formamide, 10% DMSO, 2× SCC, 2 mM vanadyl ribonucleoside, 60 µg/ml tRNA, 30 µg/ml BSA plus 0.75 µg CAG8x- or CAGG8x-Cy3 DNA oligonucleotide probe (Sigma). Following FISH, the coverslips or slide were washed twice successively in 2 × SCC/50% formamide, in 2× SCC and in PBS. The coverslips or slides were incubated 2 h with primary polyclonal antibody against MBNL1 (1/100 dilution, gift of Prof. Charles Thornton) and monoclonal rbFOX1 (1/100 dilution, gift of Prof. Doug Black) or monoclonal rbFOX2 (Abcam ab57154). Slides or coverslips were washed twice with PBS before incubation with a goat anti-rabbit secondary antibody conjugated with Alexa-Fluor 488 (1/500 dilution; Fisher Scientific SA) and donkey anti mouse secondary antibody conjugated with Cy5 (1/500 dilution; Interchim SA) for 60 min; incubated for 10 min in 2 × SCC/DAPI (1/10 000 dilution) and rinsed twice in 2 × SSC before mounting in Pro-Long media (Molecular Probes). Slides were examined using a fluorescence microscope (Leica).

**Western blotting analyses.** Proteins were denatured 3 min at 95 °C, separated on 4–12% SDS-PAGE gel (Invitrogen), transferred on nitrocellulose membranes (Whatman Protan), blocked with 5% non-fat dry milk (NFM) in Tris Buffer Saline buffer (TBS), incubated with rabbit anti-rbFOX1 (Abcam ab83574), anti-MBNL1 (gift of Prof. Glenn Morris) in TBS-5% NFM, washed 3 times and incubated with anti-rabbit or -mouse Peroxidase antibody (Jackson Immunoresearch, 1:3000) 1 h in TBS-5% NFM, followed by autoradiography using the ECL chemoluminescence system (ThermoFisher).

**RT-PCR analyses.** Total RNA from cells or patient muscle was isolated by TriReagent (Molecular Research Center). cDNAs were generated using the Transcriptor High Fidelity cDNA synthesis kit (Roche Diagnostics) for quantification of mRNAs. PCR was performed with Taq polymerase (Roche), one denaturation step at 94 °C for 2 min, 30 cycles of amplification 94 °C for 1 min, 60 °C for 1 min, 72 °C for 2 min and a final step at 72 °C for 5 min using the primer described below. The PCR products were precipitated, analyzed by electrophoresis on a 6.5% poly-acrylamide gel, stained with ethidium bromide and quantified with a Typhoon scanner.

**Oligonucleotides.**

MINIGENE SPLICING–RT-PCR
INSR-FW TAATACGACTCACTATAGGGC
INSR-REV GCTGCAATAAACAAGTTCTGC
ATP5C1-FW GTCATCACAAAAGAGTTGATTG
ATP5C1-REV CACTGCATTCTAGTTGTGGTTTGT

ENDOGENE SPLICING–RT-PCR
Human TBX3-FW CACGGTAGAACGAGGCATTT
Human TBX3-REV AGACGGGGCTGATTAACCTT
Human ENAH-FW TGCTTCAGCCTGTCATAGTCA
Human ENAH-REV TGGCAGCAAGTCACCTGTTA
Human ECT2-FW CAGAATCCTGAAAGTCCGTGA
Human ECT2-REV TTGGTTCAAGAAGCTGGAAAA
Human CACNA1S-FW AGGAGGGTTCGCACTCCTTCT
Human CACNA1S-REV GCTACTTTGGAGACCCCTGGA
Human RYR1-FW GACAACAAAAGCAAAATGGC
Human RYR1-REV CTTGGTGCGTTCCTGGTCCG
Human ATP2A1-FW CCATCGGTGCATGCCGAACGAGC
Human ATP2A1-REV CTCGTGGGCTCCATCTGCCTGTCC
Human LDB3-FW GCAAGACCCTGATGAAGAAGCTC
Human LDB3-REV GACAGAAGGCCGGATGCTG
Human MBNL1-FW GCTGCCCAATACCAGGTCAAC
Human MBNL1-REV TGGTGGGAGAAATGCTGTATG

ENDOGENE EXPRESSION–qRT-PCR
Human MBNL1 FW CATTTGCAAGCCAAGATCAA
Human MBNL1 REV AACTGGTGGGAGAAATGCTG
Human MBNL2 FW GCCCAGGCAGATGCAATTTAT
Human MBNL2 REV GTGGACTTCCGGGAACAATA
Human rbFOX1 FW CCCGAGCACACATTAAACCT
Human rbFOX1 REV CGGAACCTCAAGGGGATATT
Human rbFOX2 FW AAGCCCAGTAGTTGGAGCTG
Human rbFOX2 REV CAAATGGGCTCCTCTGAAAG
Human CNBP FW TGCTATAACTGCGGTAGAGGT
Human CNBP REV TTGAATGTGTCCGAATTCTCC
Human DMPK FW CATGAACAAGTGGGACATGC
Human DMPK REV CAATCTCCGCCAGGTAGAAG
Mouse Mbnl1 FW CGGGACACAAAATGGCTAAC
Mouse Mbnl1 REV TTGCAGTTCTCTCTCTGGAGCA
Mouse Mbnl2 FW TTTTCCCACATCCTCCAAAG
Mouse Mbnl2 REV GAATGTGTCAGCAAGCAGGA
Mouse rbFox1 FW GCATAGAAGTCGGGGCTGTA
Mouse rbFox1 REV GAGGGAGAAATTGCACGGTA
Mouse rbFox2 FW AACCAGGAGCCAACAACAAC
Mouse rbFox2 REV TGTCTGTGCTCCACCTTCTG
Mouse Cnbp FW TCACCTTGGTGCAGTCTTTTT
Mouse Cnbp REV ACTGCAAGGAGCCCAAGAG
Mouse Dmpk FW AGCGGTGGTGAAGATGAAAC
Mouse Dmpk REV AGTGCAGCTGTGTGATCCAG
RPLPO FW GAAGGTGTAATCCGTCTCCA
RPLPO REV GAAGTCACTGTGCCAGCCCA

DROSOPHILA TRANSGENES EXPRESSION - RT-PCR and qRT-PCR
SV40 FW GGAAAGTCCTTGGGGTCTTC
SV40 REV GGAACTGATGAATGGGAGCA
Drosophila RP49 FW ATGACCATCCGCCCCAGCATAC
Drosophila RP49 REV ATGTGGCGGGTGCGCTTGTTC
Drosophila Fhos FWD GTCATGGAGTCGAGCAGTGA
Drosophila Fhos REV TGTGATGCGGGTATCTACGA
Drosophila Serca FWD GCAGATGTTCCTGATGTGCG

Drosophila Serca REV CGTCCTCCTTCACATTCAC

GEL SHIFT SEQUENCES
Vector sequence is indicated in lowercase, sequence of interest is indicated in uppercase.
UGCAUGC: gggagacccaagctggctagcgtttaaacttaagcttTGCATGCg
UGCCUGC: gggagacccaagctggctagcgtttaaacttaagcttTGCCTGCg
(CUG)Nx: gggagacccaagctggctagcgtttaaacttaagctt (CTG)Nx g
(CCUG)Nx: gggagacccaagctggctagcgtttaaacttaagctt (CCTG)Nx g

Human *INSR*
gggagacccaagctggctagcgtttaaacttaagcttGTGCGACCCCTGGTGCCTGCTCCGCG
CAGGGCCGGCGGCGTGCCAGGCAGATGCCTCGGAGAACCCAGGGGTTT
CTGTGGCTTTTTGCATGCGGCGGGCAGCTGTGCTg

Human *BIN1*
gggagacccaagctggctagcgtttaaacttaagcttTACCGGCAGTGAGTGCTGCGGAGGGG
CGCAGAGGCCCGCGCCCTGGCTGGCCCTGTGCATGCGCCTTGCGCCCTG
CTCCCAGGTGCCACTAACCCGTAATCTGGCTCTGTGTGCAGTGCTGCCC
GGCAGGGCTGTCGTGTGCGTGTTGGGTGGGAAg

Human *TNNT2*
gggagacccaagctggctagcgtttaaacttaagcttAACCACTGCGCTGGGTGGCTGCTCCTG
CCGCGGGCTCTCTGCTCCCAGACTAACCTGTCTCGCTTTTCCCCTCCGCT
GCGGCCACTCCCTGAACCTCAGAAGAGGAGGACTGGAGAGAGGACGAA
GACGGTAGTACAGCCTTTCCTTCTGTGGTGCTTTCTGCTGCCTGCTGTCC
CAAGTGCAGCCTCCTTGTCCAGGGGCCCTGTTCTGGGGGCTGGGGGGT
GTGAGTAGGCGGCAGGGACGGAGTGGGTCAGTCGTTTCCTg

Mouse *Clcn1*
gggagacccaagctggctagcgtttaaacttaagcttTCCATGTTTCCTCCTGTGCTGCCCCCG
TTCTTCTGTGCTTCCTGACACCCATCCACCTGGTTTACATACCACCTGTC
TGTCCCCCTCTGCCACCTGCCTCGCCCGTCGTGCTTCTCTGTTGCAGACC
GTGCCTGGGCAGCTTGATCTCCTGGTGCCAGCCTGTGCAGTGGGCGTGG
GATGCTACTTTGCAGCCCCTGATGGAGGCAAGTTTCACTTCCTCCCTAC
CTTGGTTGCCTGAGCCAGACTTGGAGGGGTGGTTTGTGTCTGCTGTGG
CTTTGGGGTTGAGGGGCCATGCTCACTGAAGGAACTTATGGGTGGGCg

**Data availability**. rbFOX1 RRM binding to the UGCCUGC RNA sequence was computed using rbFOX1 RRM bound to the UGCAUGC sequence, which coordinates have been previously deposited to the Protein Data Bank under the 2ERR accession code[49]. The data that support the findings of this study are available within the article and its supplementary information files or from the corresponding authors on reasonable request.

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

# ARTICLE

40. Nakahata, S. & Kawamoto, S. Tissue-dependent isoforms of mammalian Fox-1 homologs are associated with tissue-specific splicing activities. *Nucleic Acids Res.* **33**, 2078–2089 (2005).

41. Underwood, J. G., Boutz, P. L., Dougherty, J. D., Stoilov, P. & Black, D. L. Homologues of the Caenorhabditis elegans Fox-1 protein are neuronal splicing regulators in mammals. *Mol. Cell Biol.* **25**, 10005–10006 (2005).

42. Zhang, C. et al. Defining the regulatory network of the tissue-specific splicing factors Fox-1 and Fox-2. *Genes Dev.* **22**, 2550–2563 (2008).

43. Yeo, G. W. et al. An RNA code for the FOX2 splicing regulator revealed by mapping RNA-protein interactions in stem cells. *Nat. Struct. Mol. Biol.* **16**, 130–137 (2009).

44. Venables, J. P. et al. MBNL1 and RBFOX2 cooperate to establish a splicing programme involved in pluripotent stem cell differentiation. *Nat. Commun.* **4**, 2480 (2013).

45. Weyn-Vanhentenryck, S. M. et al. HITS-CLIP and integrative modeling define the Rbfox splicing-regulatory network linked to brain development and autism. *Cell Rep.* **6**, 1139–1152 (2014).

46. Damianov, A. et al. Rbfox proteins regulate splicing as part of a large multiprotein complex LASR. *Cell* **165**, 606–619 (2016).

47. Carreira-Rosario, A. et al. Repression of Pumilio protein expression by Rbfox1 promotes germ cell differentiation. *Dev. Cell.* **36**, 562–571 (2016).

48. Kim, K. K., Adelstein, R. S. & Kawamoto, S. Identification of neuronal nuclei (NeuN) as Fox-3, a new member of the Fox-1 gene family of splicing factors. *J. Biol. Chem.* **284**, 31052–31061 (2009).

49. Auweter, S. D. et al. Molecular basis of RNA recognition by the human alternative splicing factor Fox-1. *EMBO J.* **25**, 163–173 (2006).

50. Singh, R. K. et al. Rbfox2-coordinated alternative splicing of Mef2d and Rock2 controls myoblast fusion during myogenesis. *Mol. Cell.* **55**, 592–603 (2014).

51. Wei, C, et al. Repression of the central splicing regulator RBFox2 is functionally linked to pressure overload-induced heart failure. *Cell Rep.* **10**, 1521-1533 (2015).

52. Tran, H. et al. Analysis of exonic regions involved in nuclear localization, splicing activity, and dimerization of Muscleblind-like-1 isoforms. *J. Biol. Chem.* **286**, 16435–16446 (2011).

53. Salisbury, E. et al. Expression of RNA CCUG repeats dysregulates translation and degradation of proteins in myotonic dystrophy 2 patients. *Am. J. Pathol.* **175**, 748–762 (2009).

54. Jones, K. et al. RNA Foci, CUGBP1, and ZNF9 are the primary targets of the mutant CUG and CCUG repeats expanded in myotonic dystrophies type 1 and type 2. *Am. J. Pathol.* **179**, 2475–2489 (2011).

55. Cardani, R. et al. Overexpression of CUGBP1 in skeletal muscle from adult classic myotonic dystrophy type 1 but not from myotonic dystrophy type 2. *PLoS One* **8**, e83777 (2013).

56. Zu, T. et al. Non-ATG-initiated translation directed by microsatellite expansions. *Proc. Natl Acad. Sci. USA* **108**, 260–265 (2011).

57. Zu, T. et al. RAN translation regulated by Muscleblind proteins in myotonic dystrophy type 2. *Neuron* **95**, 1292–1305.e5 (2017).

58. Kino, Y. et al. Muscleblind protein, MBNL1/EXP, binds specifically to CHHG repeats. *Hum. Mol. Genet* **13**, 495–507 (2004).

59. Warf, M. B. & Berglund, J. A. MBNL binds similar RNA structures in the CUG repeats of myotonic dystrophy and its pre-mRNA substrate cardiac troponin T. *RNA* **13**, 2238–2251 (2007).

60. Yuan, Y. et al. Muscleblind-like 1 interacts with RNA hairpins in splicing target and pathogenic RNAs. *Nucleic Acids Res.* **35**, 5474–5486 (2007).

61. Phair, R. D. & Misteli, T. High mobility of proteins in the mammalian cell nucleus. *Nature* **404**, 604–609 (2000).

62. Kruhlak, M. J. et al. Reduced mobility of the alternate splicing factor (ASF) through the nucleoplasm and steady state speckle compartments. *J. Cell. Biol.* **150**, 41–51 (2000).

63. Gehman, L. T. et al. The splicing regulator Rbfox1 (A2BP1) controls neuronal excitation in the mammalian brain. *Nat. Genet.* **43**, 706–711 (2011).

64. Gehman, L. T. et al. The splicing regulator Rbfox2 is required for both cerebellar development and mature motor function. *Genes Dev.* **26**, 445–460 (2012).

65. Pedrotti, S. et al. The RNA-binding protein Rbfox1 regulates splicing required for skeletal muscle structure and function. *Hum. Mol. Genet.* **24**, 2360–2374 (2015).

66. Nutter, C. A. et al. Dysregulation of RBFOX2 is an early event in cardiac pathogenesis of diabetes. *Cell Rep.* **15**, 2200–2213 (2016).

67. Mankodi, A. et al. Myotonic dystrophy in transgenic mice expressing an expanded CUG repeat. *Science* **289**, 1769–1773 (2000).

68. Lee, J. A. et al. Cytoplasmic Rbfox1 regulates the expression of synaptic and autism-related genes. *Neuron* **89**, 113–128 (2016).

69. Bhalla, K. et al. The de novo chromosome 16 translocations of two patients with abnormal phenotypes (mental retardation and epilepsy) disrupt the A2BP1 gene. *J. Hum. Genet.* **49**, 308–311 (2004).

70. Davis, L. K. et al. Rare inherited A2BP1 deletion in a proband with autism and developmental hemiparesis. *Am. J. Med. Genet. A* **158A**, 1654–1661 (2012).

71. Lal, D. et al. Extending the phenotypic spectrum of RBFOX1 deletions: Sporadic focal epilepsy. *Epilepsia* **56**, e129–e133 (2015).

72. Homsy, J. et al. De novo mutations in congenital heart disease with neurodevelopmental and other congenital anomalies. *Science* **350**, 1262–1266 (2015).

73. McKean, D. M. et al. Loss of RNA expression and allele-specific expression associated with congenital heart disease. *Nat. Commun.* **7**, 12824 (2016).

74. Verma, S. K. et al. Rbfox2 function in RNA metabolism is impaired in hypoplastic left heart syndrome patient hearts. *Sci. Rep.* **6**, 30896 (2016).

75. Wheeler, T. M. et al. Reversal of RNA dominance by displacement of protein sequestered on triplet repeat RNA. *Science* **325**, 336–369 (2009). Jul 17.

76. Warf, M. B., Nakamori, M., Matthys, C. M., Thornton, C. A. & Berglund, J. A. Pentamidine reverses the splicing defects associated with myotonic dystrophy. *Proc. Natl Acad. Sci. USA* **106**, 18551–18556 (2009).

77. Konieczny, P. et al. Myotonic dystrophy: candidate small molecule therapeutics. *Drug. Discov. Today* **22**, 1740–1748 (2017).

78. Garcia-Lopez, A. et al. Genetic and chemical modifiers of a CUG toxicity model in Drosophila. *PLoS One* **3**, e1595 (2008).

79. Ordway and Detloff. In vitro synthesis and cloning of long CAG repeats. *Biotechniques* **21**, 609–610 (1996).

## Acknowledgements

We thank Tom Cooper for the gift of the DT960 vector, RTB300 and *INSR* exon 11 minigenes, Kunio Inoue for the gift of the human mitochondrial ATP synthase gamma-subunit exon 9 minigene (F1gamma), Christiane Branlant and Maury Swanson for the gift of the pGEX-MBNL1-Δ101 and PGEX-MBNL1-HIS vectors, Douglas Black for the gift of the rbFOX1 and rbFOX2 expression plasmids and the kind gift of monoclonal anti-rbFOX1 antibody, Glenn Morris for the gift of the anti-MBNL1 monoclonal antibody, Charles Thornton for the gift of the anti-MBNL1 polyclonal antibody. This work was supported by ERC-2012-StG #310659 "RNA DISEASES"; AFM #18833 "Model DM" (NCB), ANR-10-LABX-0030-INRT (IGBMC); ANR-10-IDEX-0002-02 (IGBMC); ERARE-12-059 "HEART DM" (NCB, BL, DF), PrometeoII (BL), Fondazione Malattie Miotoniche (GM), UPMC/INSERM/CNRS/AIM (DF), Research Resource Network Japan (HF), grants-in-aids from JSPS (MPT), Intramural Research Grant of NCNP 26-8 (MPT) and research Grants for Intractable Disease from the Ministry of Health, Labor and Welfare (MPT).

## Author contributions

Experiments were performed by C.S., E.C., M.B., F.F., A.G., F.R., and P.S. Clinical samples and patient data were obtained from J.P., B.U., J.W.D., G.M., G.B., H.F., M.T., B. S., and D.F. Data were collected and analyzed by C.S., R.A., F.A., B.L., and N.C.-B. The study was designed and coordinated by C.S., B.L., and N.C.-B.
