## [Peer Review File · Nature Communications]

Reviewers' comments:

Reviewer #1 (Remarks to the Author):

The paper by Sellier et al presents the series of very interesting experimental data suggesting protective effect of RBFOXs on pathomechanism of myotonic dystrophy type 2 (DM2) caused by accumulation of toxic RNA having CCUG expansions (CCUGexp). This toxic RNA similarly to CUG expansion (CUGexp), which causes DM1, sequester MBNL proteins in cell nuclei leading to abnormalities in metabolism of hundreds of other transcripts. The main achievement of research is demonstration of competition of RBFOXs and MBNLs in binding to CCUGexp but not CUGexp in vitro and in cellular models. General findings are novel, manuscript shows many interesting data important to others, not only from DM field, and uses several complementary methods, however, needs some additional experimental data, information about current knowledge about differences between DM1 and DM2 and some explanations.

Major points:

Intro: Authors presented that the length of CCUG is generally higher than CUG repeat tract and that expression level of mature mutant mRNAs is higher in DM2 than DM1 what not correspond with more severe clinical phenotype of DM1, but they did not show that probably the most important difference between mutation repeats in DM1 and DM2 is location of expanded sequence. CUGexp are located in 3'UTR of mature mRNA which is much stable in cells while CCUGexp are in intron which is mainly excised from pre-mRNA and its stability is probably much lower. The same info should be included into Discussion section.

Intro: It should be clarified that in Drosophila DM1 and DM2 models where similar phenotypes were observed have not natural sequence context of mutant genes and different localization of repeats in RNAs compare to humans.

Intro: Abnormalities in the level of CELF1 protein, especially concerning its phosphorylation status, were not clearly shown in DM2 and was only shown in the most affected DM1 patients.

Fig. 2: It is hard to compare results obtained in two different biochemical studies, what Authors did in Results and Discussion, giving exact number of differences of Kds for different RNAs. It is needed to show direct comparison of binding affinity of RBFOX1 to CCUG repeats and to other natural RBFOX targets. Similarities in Kd values for MBNL and RBFOX obtained in in vitro studies are not translated directly into in vivo situation. Different purification products of the same recombinant protein can often give significant differences in calculated Kds (sometimes even 10-times).

Fig. 4: It should be clarified how Authors calculated the Dendra2-fusion protein signals (shown in A and B) for control experiments such as rbFOX1+CUG960 or rbFOX1. It is not clear why there is such a long time of fluorescence rescue in control experiments and from which spots they measure fluorescence recovery in rbFOX1+CUG960x cells. One could expect the t1/2 in rather few seconds range but not over 1 minute as we can read from graphs. Statistical analysis of obtained results is also needed. It looks like the curve for rbFOX1+CCUG1000x spots would be significantly different than for control spots.

Fig. 5: The Expression level of MBNL pool (1, 2, 3) and RBFOX pool (1, 2) in skeletal muscles and muscle cells is different. Authors used all these models in experiments and would be good to see the differences. Previously published or publically available results of whole-transcriptome studies are showing that the quantity of MBNLs pool is 2-5-times higher than the pool of RBFOX mRNAs in skeletal muscles. The question is whether Author's statement suggesting that "... rbFOX proteins, by competing with and reducing the titration of MBNL1 within CCUG RNA foci, may participate to the lesser toxicity of CCTG repeats in myotonic dystrophy type 2" would be correct for

physiological conditions, especially if affinity of RBFOXs to CCUG is 20-times lower than to known targets. Moreover, to check the effect of RBFOX on splicing of MBNL-dependent exons only overexpression experiments were shown (Fig. 5D). We could expect that silencing of RBFOX pool in cells with overexpression of CCUG can shift splicing profile of CLCN1 or TNNT3 toward a profile specific for cells with overexpression of CUG repeats. This experiments should be done. Also silencing of RBFOX pool should change the foci status toward more DM1-specific (Fig B is showing only the results for mass RBFOX1 overexpression).

Fig 6: Authors claimed that "...expression of rbFOX1 corrects alternative splicing misregulations caused by titration of MBNL1 by expanded CCUG repeats": here better to use overexpression instead of expression. However, correction of splicing of selected MBNL-sensitive exons in flies would be caused directly by overexpression of RBFOX (many exons are regulated by these two classes of proteins): *Serca1* and *Fhos*. Control experiments are needed. Observed effects (molecular, behavioral and physiological) of RBFOX1 would be caused by overexpression of RBFOX1 protein itself. The model with overexpression of RBFOX1 w/o repeats (CUG CCUG) in flies would be a good experimental control to solve this issue. Probably Authors have such model.

Minor points:

Correct the sentence: "We controlled that rbFox1, but not Mbnl1, regulates the splicing of these exons (Supplementary Figures 4B and 4C)."

Correct the name of *Tbx3* and *Enah* endogens on Fig. S4.

The sentence "However, we found that the binding of rbFOX1 to ten repeated UGCCUGC motifs was much lower than the binding to a single UGCAUGC motif, and that the binding of rbFOX1 to a single UGCCUGC motif was barely detectable." is unclear for me.

Reviewer #2 (Remarks to the Author):

To address the mechanisms by which myotonic dystrophy type 2 (DM2) manifests less severe than myotonic dysrophy type 1 (DM1), Sellier et al group report that the RNA binding protein (RBP) RBFOX may compete with MBNL binding sites for DM2-related CCUG repeats freeing up MBNL for its normal function. The authors' following findings are compelling evidence in the favor of this mechanism:

1. UV-crosslinking experiment combined with Mass spectrometry shows that both MBNL and RBFOX proteins associate with biotinylated CCUG repeats but only MBNL proteins in the nuclear extract associate with CUG repeats (Figure 1).
2. Rbfox colocalizes with CCUG but not CUG foci in transfected as well as primary patient cells and tissues. The authors tested colocalization for many different RBPs and showed that RBFOX specifically colocalize with CCUG repeats (Figure 2, 3, and supplementary figure 1).
3. RBFOX1 associates with CCUG repeats in vitro as shown by electrophoretic mobility shift assays (EMSAs, Figure 2)
4. It's clear that RBFOX function is not lost in cells expressing CCUG repeats or in DM2 (Figure 4) as known RBFOX regulated alternative splicing (AS) events are not altered in cells expressing CCUG or in DM2.
5. In vitro UV-crosslinking and competition assay shows that RBFOX1 may compete off MBNL1

protein from CCUG repeats (Figure 5 A). Furthermore, the authors show that overexpressing RBFOX1 reduces MBNL colocalization with CCUG repeats in primary cells and cells transfected with CCUG repeats concluding that RBFOX1 displaces MBNL from the CCUG foci.

6. RBFOX1 modulates AS events regulated by MBNL1.

7. Lastly, Rbfox1 expression in CCUG but not CUG flies prevents muscle loss and prolongs survival.

The study is well performed, experiments are clear, and well organized and establishes the role of RBFOX1 as a modifier gene. However, some doubts remain about the mechanism as noted below:

1. Although RBFOX1 binds CCUG repeats *in vitro*, it's not clear if it can bind and displace MBNL *in vivo*. The cellular experiments where RBFOX1 expression decreases MBNL1 colocalization with CCUG repeats could be attributed to increases in MBNL (MBNL1 or MBNL2) expression or change in their isoforms. RBFOX and MBNL proteins are both known to regulate AS of each other. The authors can provide isoform analysis and western blots of all MBNL and RBFOX proteins as controls in all systems used but I understand that can be tedious (see below for experimental suggestions).

2. Furthermore, RBFOX binding sites do exist in TNNT2 (cTNT) and CLCN1 splicing rescue is not impressive and may be attributed to effects of RBFOX on minigenes itself. Authors can perform UV-crosslinking to show that RBFOX does not bind these minigenes.

3. Since MBNL is not diffusible and sticks to the CCUG repeats, it is somewhat difficult to understand how physiological levels of RBFOX can displace it and still maintain its function.

Experiment suggestions:

1. To answer all these questions and not do an exhausting amount of control experiments, the authors can perform CLIP-seq on RBFOX and MBNL in DM2 cells and tissues. This will easily answer the question if RBFOX1 directly binds CCUG repeats. This assay was reported by Goodwin et al., 2015 and published in *Cell Reports* where the authors showed plethora of MBNL binding to CUG and CCUG repeats in disease tissue. If authors would like, they can also do RBFOX CLIP-seq in cells expressing CCUG repeats where MBNL1 is knocked down to see RBFOX binding in that system. In absence of these experiments, it's hard to reason RBFOX displacing MBNL from its target sites while still maintaining its function especially when it accurately splices its own targets. This experiment will obviate the necessity of many many other controls.

2. Unlike what authors mention, Rbfox1 (Gehman et al from Doug Black's group) and Mbnl2 (Charizanis et al from Maurice Swanson's group) do share many similar targets in the brain. So it's quite possible that RBFOX1 is a modifier for MBNL related effects, which is still an interesting finding but the authors argue against it by showing that RBFOX does not affect CUG repeat containing models. CLIP-seq (or HITS-CLIP) on both RBFOX and MBNL in WT and CCUG repeat containing models will also provide an overlap of RNA targets of these two RBPs. Essentially CLIP-seq for MBNL1, MBNL2, and RBFOX1 on DM2 tissues and primary cells will clarify all doubts and answer key questions about the mechanism of RBFOX-related modulation of DM2 related molecular pathology and phenotypes.

3. It will also be good to know if MBNL and RBFOX proteins in an RNA dependent or independent manner to further clarify mechanism of co-regulation although not necessary especially if CLIP-seq is done.

Reviewer #3 (Remarks to the Author):

This manuscript demonstrates a surprising ability of Rbfox proteins to bind UUGC repeats and thereby ameliorate the phenotype of DM2 patients. Substantial data supports a model in which Rbfox proteins compete with MBNL1 for binding UUGC repeats so as to reduce sequestration of MBNL1. The authors make a strong case that a UUGCUUGC repeat, which resembles the Rbfox binding site (UGCAUG) at five of six positions, binds Rbfox proteins with reduced but sufficient affinity to that Rbfox proteins co-localize with nuclear foci of repeat RNA. Remarkably, the aberrant splicing that is normally observed in the presence of UUGC repeats is partially corrected by Rbfox over-expression, and the phenotype of DM2-model *Drosophila* is ameliorated as well.

This paper makes a major contribution to our understanding of differences in phenotype of patients with DM1 vs DM2. The study is very well done and I have only a few minor questions.

1. p. 6: It is not clear how mass spectrometry alone could identify the ~35 kDa protein as Rbfox proteins (Supplementary Tables 1 and 2). Are these the only proteins of this size that are detected in association with UUGC but not UGC repeats?

2. Figure 1E: the inset figures for the merged images do not seem to show a significant difference between Rbfox colocalization with CUG vs CCUG repeats. (However, the overall Rbfox1 signal seems substantially more focal in CCUG cells and aligned with the RNA signals.)

3. Figure 4. Maybe this is a semantic issue, but the data in 4A show loss of photoconverted dendra-rbFox1 immunofluorescence signal with time, but they do not directly show that the signal diffuses away into the nucleoplasm. Admittedly, the alternative explanation that the signal decays rather than diffuses is unlikely, given that the signal of dendra-MBNL1 is barely reduced in the same time frame).

4. lines 493-5: what is the data supporting the statement that rbFOX1 binds better to one UGCAUGC motif than to ten repeated UGCCUGC motifs, and that the binding of rbFOX1 to a single UGCCUGC motif was barely detectable?

Minor issues

1. Figure legend 2E: says magnification of (F) but should be (D). Show image for binding UGCAUG??

2. The text is inconsistent in its use of the terms supplemental vs supplementary figures and tables.

3. While the manuscript is very well written, there are a few spelling mistakes that should be carefully edited. A few items that I noticed include the following: line 319: dependent is misspelled; figure 5 legend: transfected is misspelled; Figure 6C: the word height is misspelled on the Y axis; line 514: neurodegenerative is misspelled.

Reviewer #4 (Remarks to the Author):

NCOMMS_17_21823

Sellier et al. investigated the role of rbFOX proteins in DM2 pathogenesis. The authors showed that rbFOX1 interacts directly with CCUG RNA, and that rbFOX co-localizes with MBNL1 in CCUG RNA foci of DM2. Interestingly, rbFOX1 overexpression can rescue CCUG RNA-mediated splicing defects

and toxicity. The authors propose that rbFOX protein competes with the binding of MBNL1 for CCUG RNA and displaces sequestered MBNL1 protein from RNA foci. The findings are interesting and would bring new insights into the mechanistic understanding of DM2 pathogenesis. However, there are several points that require further clarification and verification.

Major comments:

It is not clear whether rbFOX1 is a short CCUG RNA-binding protein rather than an expanded CCUG RNA binder. Given that the CCUG 3x sequence already mirrors an endogenous rbFOX binding site, it is reasonable to speculate that rbFOX1 is an unexpanded r(CCUG)₃-binding protein.

Does the endogenous UGCAUGC sequence compete with CCUG for rbFOX1 in gel shift assay and vice versa?

It is not certain how the authors may relate the unexpanded rbFOX1/CCUG 10X binding to CCUG RNA foci formation, the latter requires the presence of up to hundreds to a thousand copies of CCUG to form. May rbFOX1 suppress DM2 through multiple distinct mechanisms?

Can MBNL1 displace rbFOX1? Do other rbFOX proteins have the same ability in competing with MBNL1 when overexpressed?

The authors assume that the binding mechanism of rbFOXs and MBNL1 to CCUG RNA is mechanistically similar to each other. If so, why would the interaction of MBNL1 to CCUG result in sequestration, while the rbFOX/CCUG RNA interaction doesn't cause rbFOX immobilization?

Does the increased level of rbFOX1 expression in C2C12 cells result in a more intense CCUG foci co-localization, and at the same time reduce the intensity of foci-sequestered MBNL1?

What is the binding affinity of rbFOX1 to CCUG 20X?

In Fig. 2A, CCUG 10X was defined as expanded CCUG repeat. 10 repeat is not considered as expanded. The authors used CCUG 10x and 30x RNAs in the different gel shift experiments. Did they observe CCUG length-dependent binding of rbFOX1?

Page 7 line 197: "All other tested candidate proteins do not co-localize significantly with CUG or CCUG RNA foci (Figure 1F and Supplemental figure 1)."

Would these proteins interact with shorter repeat RNA sequences, e.g. in a gel-shift assay?

What is the rbFOX1 ortholog in *Drosophila*? Should a displacement hypothesis be tested in *Drosophila*, it is more appropriate to use the *Drosophila* rbFOX1 protein in the overexpression experiment. This is because the MBNL1 counterpart would be endogenous fly MBNL1.

Page 13 lines 425-426 "Importantly, expression of rbFOX1 fully corrects the splicing mis-regulations of Fhos and Serca mRNAs caused by expression of expanded CCUG repeats in DM2 flies (Figures 6E and 6F)."

Given the RNA-binding property of rbFOX1, this correction may be achieved through mechanism(s) together with/other than the MBNL1 displacement hypothesis. The authors should further explore this.

To substantiate the competition/displacement hypothesis, it should further be tested in cell-free system(s) using expanded repeat sequences that correspond to the cell/animal models.

Minor comments:

What are sequence similarities between the 3 human rbFOX proteins and the Drosophila ortholog?

Page 9 line 267: It is not clear how many pairs of DM1/DM2 patient samples were presented in this manuscript.

Page 9 line 269: Figure 3C was mislabeled as 3H in figure legend.

Page 10 line 313: We "concluded" that rbFOX1 ...

Page 10 line 322: A rbFOX1 knockdown control was missing.

Page 11 line 326: the "extent" of splicing ...

Page 11 lines 328-329: "These results indicate that, in contrast to MBNL1, the splicing regulatory functions of the rbFOX proteins are not altered by expression of expanded CCUG repeats." Could this be due to redundancy of rbFOX proteins?

Fig. 5B: A mutant rbFOX protein control would be needed to demonstrate the effect of rbFOX altering the subnuclear distribution of MBNL1.

1 000/1,000/1000 are used interchangeably in the manuscript.

Response to Reviewers' comments:

Please find below (in blue for clarity) a point-by-point response to Referees comments and suggestions. Overall, all questions have been addressed and novel data, confirming and extending our previous results, are provided. Notably, shRNA-mediated depletion of rbFOX1 demonstrates increased localization of MBNL1 in CCUG RNA foci in primary cultures of human DM2 patient muscle cells. We also added various controls that were requested by the Referees. These novel data show that rbFOX1 overexpression does not alter MBNL1 expression, localization or regulatory function in our different cell systems. Furthermore, we added various novel gel shift experiments of MBNL1 and rbFOX1 binding to CUG, CCUG and to their normal RNA targets. However, we did not perform the CLIP-RNA Seq of rbFOX1 and MBNL1 to study their splicing co-regulation in tissue of individuals with DM2 as suggested by Referee #2. The reason is that more than 15 mg of patient tissue is required for these assays and we simply do not have access to such amount of human biological material. Furthermore, we believe that while the co-regulation of mRNAs targets by rbFOX1 and MBNL1 is an exciting research topic, it is beyond the scope of the present article that focus on the protective role of rbFOX1 in myotonic dystrophy type 2. We hope that these modifications will be adequate for publication in *Nature Communications*. Also, we would like to thank you and all four Referees for their comments and suggestions that helped us to obtain a better and clearer manuscript.

REVIEWERS' COMMENTS:

REVIEWER #1:

The paper by Sellier et al presents the series of very interesting experimental data suggesting protective effect of RBFOXs on pathomechanism of myotonic dystrophy type 2 (DM2) caused by accumulation of toxic RNA having CCUG expansions (CCUGexp). This toxic RNA similarly to CUG expansion (CUGexp), which causes DM1, sequester MBNL proteins in cell nuclei leading to abnormalities in metabolism of hundreds of other transcripts. The main achievement of research is demonstration of competition of RBFOXs and MBNLs in binding to CCUGexp but not CUGexp in vitro and in cellular models. General findings are novel, manuscript shows many interesting data important to others, not only from DM field, and uses several complementary methods, however, needs some additional experimental data, information about current knowledge about differences between DM1 and DM2 and some explanations.

Major points:

1. Intro: Authors presented that the length of CCUG is generally higher than CUG repeat tract and that expression level of mature mutant mRNAs is higher in DM2 than DM1 what not correspond with more severe clinical phenotype of DM1, but they did not show that probably the most important difference between mutation repeats in DM1 and DM2 is location of expanded sequence. CUGexp are located in 3'UTR of mature mRNA which is much stable

in cells while CCUGexp are in intron which is mainly excised from pre-mRNA and its stability is probably much lower. The same info should be included into Discussion section.

Indeed, the genomic localization of expanded CCUG and CUG repeats differs with the CCUG repeats located within the first intron of the *CNBP* pre-mRNA, while the CUG repeats are located in the 3'UTR of the *DMPK* mRNA. These differences are likely to influence their RNA stability and thus their expression and toxicity. The introduction and discussion have been modified accordingly.

2. Intro: It should be clarified that in *Drosophila* DM1 and DM2 models where similar phenotypes were observed have not natural sequence context of mutant genes and different localization of repeats in RNAs compare to humans.

It is now clarified that the *Drosophila* CTG or CCTG transgenes are deprived of any *DMPK* or *CNBP* natural sequences, thus reinforcing the importance of the expansion genomic localization.

3. Intro: Abnormalities in the level of CELF1 protein, especially concerning its phosphorylation status, were not clearly shown in DM2 and was only shown in the most affected DM1 patients.

These information are now added in the introduction as well as in the discussion.

4. Fig. 2: It is hard to compare results obtained in two different biochemical studies, what Authors did in Results and Discussion, giving exact number of differences of Kds for different RNAs. It is needed to show direct comparison of binding affinity of RBFOX1 to CCUG repeats and to other natural RBFOX targets. Similarities in Kd values for MBNL and RBFOX obtained in in vitro studies are not translated directly into in vivo situation. Different purification products of the same recombinant protein can often give significant differences in calculated Kds (sometimes even 10-times).

Gel-shifts assays showing binding of rbFOX1 to its natural UGCAUGY sequence are presented in a novel supplementary figure 2. We also added binding of rbFOX1 and MBNL1 to different size of CCUG repeats (10, 30 and 100 x). Similarly, novel gel-shift of recombinant purified MBNL1 binding to its natural UGC-RNA motifs containing targets (*BIN1*, *INSR*, *TNNT2* and *CLCN1*) are shown with their corresponding apparent KD in supplementary figure 2. These novel data confirm that binding of MBNL1 and rbFOX1 to 10, 30 or 100 CCUG repeats occur with similar apparent KD, suggesting that competition between these two proteins to bind to a CCUG repeats RNA is possible. Importantly, MBNL1 binds to ten CCUG repeats with a similar or higher affinity compared to its natural mRNA targets, hence supporting a titration mechanism of MBNL1 away from its normal targets by expanded CCUG repeats. In contrast, binding of rbFOX1 to ten CCUG repeats is weaker compared to its natural mRNA targets or to the UGCAUGY RNA motif, suggesting that rbFOX1 is less prone to be functionally titrated away from its normal mRNA targets by expanded CCUG repeats.

5. Fig. 4: It should be clarified how Authors calculated the Dendra2-fusion protein signals (shown in A and B) for control experiments such as rbFOX1+CUG960 or rbFOX1. It is not clear why there is such a long time of fluorescence rescue in control experiments and from which spots they measure fluorescence recovery in rbFOX1+CUG960x cells. One could

expect the $t_{1/2}$ in rather few seconds range but not over 1 minute as we can read from graphs. Statistical analysis of obtained results is also needed. It looks like the curve for rbFOX1+CCUG1000x spots would be significantly different than for control spots.

In absence of foci of Dendra2-tagged protein (for example Dendra2-MBNL1 or Dendra2-rbFOX1 in control conditions or Dendra2-rbFOX1 expressed with expanded CUG repeats), a spot of $\sim 1 \mu\text{m}$ diameter of Dendra2 tagged protein randomly chosen within the nucleoplasm was photoconverted. The method section is now clarified.

The movement of free MBNL1 is indeed in second, as in absence of expanded CUG or CCUG RNA half of photoconverted Dendra2-MBNL1 moves away from its precedent nucleoplasmic localization in less than 30 seconds. In contrast, we noted that rbFOX1 moves with a kinetics at least four times slower as it requires 2 minutes or more for half of photoconverted rbFOX1 to moves away from its initial nucleoplasmic localization. We have no clear explanation for slower movement of rbFOX1. It could be a bias caused by the Dendra2 tag or it may reflect the different mode of RNA binding of these proteins, as rbFOX1 is known to assemble in large protein complexes (Damianov et al., 2016) and form hydrogels (Ying et al., 2017). This is now specified in the discussion.

Finally, statistical analyses are now included in Figures 4A and 4B, and indicate that indeed rbFOX1 movements are slightly delayed by the expanded CCUG RNA foci, but in contrast to MBNL1, rbFOX1 is not immobilized within CCUG RNA foci. The text has been edited accordingly.

6. Fig. 5: The Expression level of MBNL pool (1, 2, 3) and RBFOX pool (1, 2) in skeletal muscles and muscle cells is different. Authors used all these models in experiments and would be good to see the differences. Previously published or publically available results of whole-transcriptome studies are showing that the quantity of MBNLs pool is 2-5-times higher than the pool of RBFOX mRNAs in skeletal muscles. The question is whether Author's statement suggesting that "... rbFOX proteins, by competing with and reducing the titration of MBNL1 within CCUG RNA foci, may participate to the lesser toxicity of CCTG repeats in myotonic dystrophy type 2" would be correct for physiological conditions, especially if affinity of RBFOXs to CCUG is 20-times lower than to known targets. Moreover, to check the effect of RBFOX on splicing of MBNL-dependent exons only overexpression experiments were shown (Fig. 5D). We could expect that silencing of RBFOX pool in cells with overexpression of CCUG can shift splicing profile of CLCN1 or TNNT3 toward a profile specific for cells with overexpression of CUG repeats. This experiment should be done. Also silencing of RBFOX pool should change the foci status toward more DM1-specific (Fig B is showing only the results for mass RBFOX1 overexpression).

Indeed, databases such as GTEx (<https://www.gtexportal.org>), BioGPS (<http://biogps.org>) or the human protein atlas (<https://www.proteinatlas.org>) consistently indicate that in human skeletal muscles MBNL1 mRNA expression is higher compared to rbFOX1 mRNA (*MBNL1*: 50 - 70 TPM, *rbFOX1*: 30 – 40 TPM), while *MBNL2* and *rbFOX2* are expressed at the same level. To confirm these results at the protein level, we performed western blotting but found that rbFOX1 and MBNL1 are expressed at roughly similar levels in adult human skeletal muscle samples. To avoid a bias due to antibody differences, recombinant purified proteins were used as standards. Quantifications yielded a concentration of MBNL1 and rbFOX1 in a similar range of ~ 1 to 2 femtomol of rbFOX1 or MBNL1 per microgram of adult human skeletal muscle (supplementary figures 5F and 5G). These results suggest that in

physiological conditions, sufficient amount of rbFOX1 is expressed to potentially compete with MBNL1.

Next, we added novel experiments showing that depletion of rbFOX1 by siRNA enhances *CLCN1* and *cTNT* splicing alterations induced by expanded CCUG repeats (Figure 6 and supplementary figure 6). As controls, depletion of rbFOX1 has no effect on splicing in CUG-expressing cells (supplementary figure 6).

Finally, we added experiments showing that shRNA-depletion of rbFOX1 increases the localization of MBNL1 within RNA foci of expanded CCUG repeats in primary cultures of muscle cells originating from individuals with DM2 (supplementary figure 5B). As controls, depletion of rbFOX1 has no effect on MBNL1 localization in DM1 muscle primary cultures (supplementary figure 5C).

7. Fig 6: Authors claimed that “...expression of rbFOX1 corrects alternative splicing misregulations caused by titration of MBNL1 by expanded CCUG repeats”: here better to use overexpression instead of expression. However, correction of splicing of selected MBNL-sensitive exons in flies would be caused directly by overexpression of RBFOX (many exons are regulated by these two classes of proteins): *Serca1* and *Fhos*. Control experiments are needed. Observed effects (molecular, behavioral and physiological) of RBFOX1 would be caused by overexpression of RBFOX1 protein itself. The model with overexpression of RBFOX1 w/o repeats (CUG CCUG) in flies would be a good experimental control to solve this issue. Probably Authors have such model.

The text has been corrected and rbFOX1 overexpression is now clearly stated and novel data showing that *Fhos* and *Serca* splicing are not altered upon the sole overexpression of rbFOX1 in *Drosophila* are now presented (Supplementary figures 7C and 7D).

Minor points:

Correct the sentence: “We controlled that rbFox1, but not Mbnl1, regulates the splicing of these exons (Supplementary Figures 4B and 4C).”

Correct the name of *Tbx3* and *Enah* endogens on Fig. S4.

The sentence “However, we found that the binding of rbFOX1 to ten repeated UGCCUGC motifs was much lower than the binding to a single UGCAUGC motif, and that the binding of rbFOX1 to a single UGCCUGC motif was barely detectable.” is unclear for me.

The text has been corrected and we would like to thank the referee for his/ her help.

REVIEWER #2:

To address the mechanisms by which myotonic dystrophy type 2 (DM2) manifests less severe than myotonic dystrophy type 1 (DM1), Sellier et al group report that the RNA binding protein (RBP) RBFOX may compete with MBNL binding sites for DM2-related CCUG repeats freeing up MBNL for its normal function. The authors' following findings are compelling evidence in the favor of this mechanism:

- UV-crosslinking experiment combined with Mass spectrometry shows that both MBNL and RBFOX proteins associate with biotinylated CCUG repeats but only MBNL proteins in the nuclear extract associate with CUG repeats (Figure 1).
- Rbfox colocalizes with CCUG but not CUG foci in transfected as well as primary patient cells and tissues. The authors tested colocalization for many different RBPs and showed that RBFOX specifically colocalize with CCUG repeats (Figure 2, 3, and supplementary figure 1).
- RBFOX1 associates with CCUG repeats in vitro as shown by electrophoretic mobility shift assays (EMSAs, Figure 2)
- It's clear that RBFOX function is not lost in cells expressing CCUG repeats or in DM2 (Figure 4) as known RBFOX regulated alternative splicing (AS) events are not altered in cells expressing CCUG or in DM2.
- In vitro UV-crosslinking and competition assay shows that RBFOX1 may compete off MBNL1 protein from CCUG repeats (Figure 5 A). Furthermore, the authors show that overexpressing RBFOX1 reduces MBNL colocalization with CCUG repeats in primary cells and cells transfected CCUG repeats concluding that RBFOX1 displaces MBNL from from the CCUG foci.
- RBFOX1 modulates AS events regulated by MBNL1.
- Lastly, Rbfox1 expression in CCUG but not CUG flies prevents muscle loss and prolongs survival.

The study is well performed, experiments are clear, and well organized and establishes the role of RBFOX1 as a modifier gene. However, some doubts remain about the mechanism as noted below:

1. Although RBFOX1 binds CCUG repeats in vitro, it is not clear if it can bind and displace MBNL in vivo. The cellular experiments where RBFOX1 expression decreases MBNL1 colocalization with CCUG repeats could be attributed to increases in MBNL (MBNL1 or MBNL2) expression or change in their isoforms. RBFOX and MBNL proteins are both known to regulate AS of each other. The authors can provide isoform analysis and western blots of all MBNL and RBFOX proteins as controls in all systems used but I understand that can be tedious (see below for experimental suggestions).

Indeed, it is known that rbFOX1 regulates *MBNL1* alternative splicing (Venables et al., 2013; Singh et al., 2014; Wei et al., 2015). To confirm and emphasize this point, we added novel RT-PCR showing that overexpression of rbFOX1 inhibits the splicing inclusion of the 36 nts exon 7 of *Mbnl1* (supplementary figure 5A). This is likely due to a rbFOX1 binding RNA motif located in the poor polypyrimidine tract of *Mbnl1* intron 6 pre-mRNA (gcaaUGCAUGaugggagGCUCAA, with rbFOX1 predicted binding site in upper case,

acceptor splice site underlined and start of MBNL1 exon 7 in upper bold case). Exon 7 is proposed to contribute to MBNL1 self-dimerization (Tran et al., 2011). Thus, overexpression of rbFOX1 in our cell systems may modify MBNL1 dimerization, hence potentially modulating MBNL1 function. However, we added novel experiments and controls showing that rbFOX1 overexpression does not modify MBNL1 expression in the various human and mouse cell systems we studied (supplementary figure 4E, supplementary figure 5E and supplementary figure 6A). Similarly, we found that rbFOX1 overexpression does not modify MBNL1 nuclear/cytoplasmic localization (Figures 5C and 5D, supplementary figures 5B and 5C). Finally, rbFOX1 overexpression has a splicing correcting effect only in CCUG expressing cells and not in control or CUG expressing cells (Figures 6A and 6B, supplementary figures 6B and 6C). Thus, it is unlikely that rbFOX1 overexpression modify the splicing regulatory functions of MBNL1.

Of interest, we also tested other alternative splicing exons of MBNL1, but rbFOX1 neither regulates alternative splicing of *Mbnl1* exon 3 (an exon of 204 nts located between the zinc fingers Znf1&2 and Znf3&4 and that regulates MBNL1 binding to RNA), nor *Mbnl1* exon 5 (a 54 nts long exon that regulates the nuclear/cytoplasmic localization of MBNL1).

Overall, these results suggest that overexpression of rbFOX1 does not correct DM2 features through a change of MBNL1 splicing. This is now discussed in the text, and we would like to thank the Referee to have bring to our attention this important potential bias.

2. Furthermore, RBFOX binding sites do exist in TNNT2 (cTNT) and CLCN1 splicing rescue is not impressive and may be attributed to effects of RBFOX on minigenes itself. Authors can perform UV-crosslinking to show that RBFOX does not binding these minigenes.

Novel gel-shift assays indicating no significant binding of rbFOX1 to *CLCN1* and cTNT (*TNNT2*) minigenes are now presented as supplementary figure 6D. As a positive control, MBNL1 binds to both *CLCN1* and cTNT RNA regulatory RNA regions.

3. Since MBNL is not diffusible and sticks to the CCUG repeats, it is somewhat difficult to understand how physiological levels of RBFOX can displace it and still maintain its function.

Indeed, the mechanisms of competition in vivo are unclear. Our tentative hypothesis is that at the moment of CCUG repeat RNA transcription, rbFOX1 binds to the nascent CCUG RNA transcript, impairing MBNL1 binding. Thus, rbFOX1 would act as a “chaperone”, protecting MBNL1 from being titrated away of its normal mRNA targets. In contrast, as MBNL1 is immobilized within CCUG RNA foci, a direct competition of rbFOX1 when the MBNL1/CCUG RNA complex is established in RNA foci is possible but less likely. The discussion has been clarified accordingly.

Experiment suggestions:

1. To answer all these questions and not do exhausting amount of control experiments, the authors can perform CLIP-seq on RBFOX and MBNL in DM2 cells and tissues. This will easily answer the question if RBFOX1 directly binds CCUG repeats. This assay was reported by Goodwin et al., 2015 and published in cell reports where the authors showed plethora of MBNL binding to CUG and CCUG repeats in disease tissue. If authors would like, they can also do RBFOX CLIP-seq in cells expressing CCUG repeats where MBNL1 is knocked down to see RBFOX binding in that system. In absence of these experiments, it's hard to reason RBFOX displacing MBNL from its target sites while still maintaining its

function especially when it accurately splices its own targets. This experiment will obviate the necessity of many many other controls.

The CLIP-RNA Seq performed in Goodwin and collaborators is impressive and the very first demonstration of a clear titration of MBNL1 away from its normal mRNA targets in human tissue samples. However, this assay was performed on human brain autopsy material using 15 mg of tissue. We do not have access to such amount of human biological material, as our experiments were performed on skeletal muscle needle biopsies.

2. Unlike what authors mention, Rbfox1 (Gehman et al from Doug Black's group) and Mbnl2 (Charizanis et al from Maurice Swanson's group) do share many similar targets in the brain. So it's quite possible that RBFOX1 is a modifier for MBNL related effects, which is still an interesting finding but the authors argue against it by showing that RBFOX does not affect CUG repeat containing models. CLIP-seq (or HITS-CLIP) on both RBFOX and MBNL in WT and CCUG repeat containing models will also provide an overlap of RNA targets of these two RBPs. Essentially CLIP-seq for MBNL1, MBNL2, and RBFOX1 on DM2 tissues and primary cells will clarify all doubts and answer key questions about the mechanism of RBFOX-related modulation of DM2 related molecular pathology and phenotypes.

Indeed, rbFOX and MBNL RNA binding proteins share and regulates numerous common mRNA targets. This co-regulation is emphasized by the proximity of their expression and the closeness of their RNA binding sites (UGC motifs for MBNL1 versus UGCAUGC for rbFOX1). However, rbFOX and MBNL proteins possess also some specific mRNA targets, which importance is highlighted by the difference of phenotype between mice knockout for either *Mbnl1* or *Mbnl2* versus mice knockout for either *rbFox1* or *rbFox2*. Similarly, loss of function mutations in the *rbFOX1* or *rbFOX2* gene in human leads to symptoms and diseases different from the one observed in myotonic dystrophy. Finally, there is no doubt that rFOX1 CLIP-RNA Seq would be a most elegant and definitive way to confirm rbFOX1 competition with MBNL1 for binding to CCUG repeats in individuals with DM2. However, we do not have access to the quantity of human material required for such assay.

3. It will also be good to know if MBNL and RBFOX proteins in an RNA dependent or independent manner to further clarify mechanism of co-regulation although not necessary especially if CLIP-seq is done.

The co-regulation of alternative splicing by rbFOX and MBNL RNA binding proteins, which is currently investigated by various groups, is exciting and of great interest. However, we believe that the mechanisms and extend of rbFOX/ MBNL splicing co-regulation is beyond the scope of the present article, which focus on the protective effect of rbFOX1 in myotonic dystrophy type 2.

REVIEWER #3:

This manuscript demonstrates a surprising ability of Rbfox proteins to bind CCUG repeats and thereby ameliorate the phenotype of DM2 patients. Substantial data supports a model in which Rbfox proteins compete with MBNL1 for binding CCUG repeats so as to reduce

sequestration of MBNL1. The authors make a strong case that a UGCCUG repeat, which resembles the Rbfox binding site (UGCAUG) at five of six positions, binds Rbfox proteins with reduced but sufficient affinity to that Rbfox proteins co-localize with nuclear foci of repeat RNA. Remarkably, the aberrant splicing that is normally observed in the presence of CCUG repeats is partially corrected by Rbfox over-expression, and the phenotype of DM2-model *Drosophila* is ameliorated as well.

This paper makes a major contribution to our understanding of differences in phenotype of patients with DM1 vs DM2. The study is very well done and I have only a few minor questions.

1. Page 6: It is not clear how mass spectrometry alone could identify the ~35 kDa protein as Rbfox proteins (Supplementary Tables 1 and 2). Are these the only proteins of this size that are detected in association with CCUG but not CUG repeats?

The text was unclear. Each protein band labelled by silver staining was cut, gel extracted and analyzed by LC-MS/MS. In the band of 35 to 40 kDa, we identified various proteins including Tra2 alpha (33 kDa), the hnRNPA1, A2 and A3 proteins (~35 to 40 kDa), TIAR (43 kDa), PCBP1 and PCBP2 (37 and 38 kDa), TDP-43 (45 kDa) and the rbFOX1, 2 and 3 proteins (35 to 42 kDa). However, only rbFOX proteins bind specifically to CCUG repeats and not to CUG repeats. The text is now clarified.

2. Figure 1E: the inset figures for the merged images do not seem to show a significant difference between Rbfox colocalization with CUG vs CCUG repeats. (However, the overall Rbfox1 signal seems substantially more focal in CCUG cells and aligned with the RNA signals.)

Better images with higher magnifications are now presented in Figures 1D and 1E.

3. Figure 4. Maybe this is a semantic issue, but the data in 4A show loss of photoconverted dendra-rbFox1 immunofluorescence signal with time, but they do not directly show that the signal diffuses away into the nucleoplasm. Admittedly, the alternative explanation that the signal decays rather than diffuses is unlikely, given that the signal of dendra-MBNL1 is barely reduced in the same time frame.

Indeed, graphs in figure 4A show the decrease of photoconverted (red) Dendra2-rbFOX1 from the laser illuminated spot. However, we concomitantly verified that the red Dendra2-rbFOX1 moves away from the laser illuminated spot to diffuse within the nucleoplasm. Thus, photoconverted Dendra2-rbFOX1 does not decay due to degradation of rbFOX1. The text is now clarified.

4. Lines 493-5: what is the data supporting the statement that rbFOX1 binds better to one UGCAUGC motif than to ten repeated UGCCUGC motifs, and that the binding of rbFOX1 to a single UGCCUGC motif was barely detectable?

Indeed, these data were missing. Gel-shift assays showing binding of rbFOX1 to a UGCAUGC RNA motif, but negligible binding to a UGCCUGC sequence, are now presented in the novel supplementary figure 2.

Minor issues

1. Figure legend 2E: says magnification of (F) but should be (D). Show image for binding UGCAUG?

2. The text is inconsistent in its use of the terms supplemental vs supplementary figures.

3. While the manuscript is very well written, there are a few spelling mistakes that should be carefully edited. A few items that I noticed include the following: line 319: dependent is misspelled; figure 5 legend: transfected is misspelled; Figure 6C: the word height is misspelled on the Y axis; line 514: neurodegenerative is misspelled.

These errors are now corrected, and we deeply thank the Referee for his/her help.

REVIEWER #4:

NCOMMS_17_21823

Sellier et al. investigated the role of rbFOX proteins in DM2 pathogenesis. The authors showed that rbFOX1 interacts directly with CCUG RNA, and that rbFOX co-localizes with MBNL1 in CCUG RNA foci of DM2. Interestingly, rbFOX1 overexpression can rescue CCUG RNA-mediated splicing defects and toxicity. The authors propose that rbFOX protein competes with the binding of MBNL1 for CCUG RNA and displaces sequestered MBNL1 protein from RNA foci. The findings are interesting and would bring new insights into the mechanistic understanding of DM2 pathogenesis. However, there are several points that require further clarification and verification.

Major comments:

1. It is not clear whether rbFOX1 is a short CCUG RNA-binding protein rather than an expanded CCUG RNA binder. Given that the CCUG 3x sequence already mirrors an endogenous rbFOX binding site, it is reasonable to speculate that rbFOX1 is an unexpanded r(CCUG)₃-binding protein.

Gel shift assays of rbFOX1 binding to a UGCCUGCC RNA motif compared to a UGCAUGC sequence with their apparent KD are now presented in supplementary figure 2. As noted previously (Auweter et al., 2006), rbFOX1 binds avidly to the UGCAUGC sequence. In contrast, rbFOX1 binds negligibly to a single UGCCUGC RNA motif (supplementary figure 2F). These data indicate that binding of rbFOX1 to CCUG repeats require expansion of that RNA motif. The text of the discussion has been modified accordingly.

2. Does the endogenous UGCAUGC sequence compete with CCUG for rbFOX1 in gel shift assay and vice versa?

Novel experiments showing competition of rbFOX1 binding to radioactively labeled ten CCUG repeats by an excess of “cold” UGCAUGY RNA and the reverse experiment are presented in the supplementary figure 2H. These data confirm that rbFOX1 binds with a higher affinity to the UGCAUGY sequence compared to CCUG repeats.

3. It is not certain how the authors may relate the unexpanded rbFOX1/ CCUG 10X binding to CCUG RNA foci formation, the latter requires the presence of up to hundreds to a thousand copies of CCUG to form. May rbFOX1 suppress DM2 through multiple distinct mechanisms?

Indeed, formation of nuclear RNA foci requires expression of at least 150 to 200 CCUG repeats in transfected cells. Specific co-localization of rbFOX1 within CCUG RNA foci is presented in the figures 1, 3, 5 and 7. As a control, rbFOX1 does not co-localize within CUG RNA foci. The gel-shift assays and the NMR structure presented in the figure 2 are demonstrating a direct binding of the rbFOX1 protein to CCUG RNA repeats of limited size (10x, 30x and 100x), but not to CUG repeats. Thus, binding of rbFOX1 to CCUG repeats *in vitro* correlates with the co-localization of rbFOX1 within CCUG RNA foci in cells. However, we do agree that *in vitro* binding is not a formal proof of the direct rbFOX1 binding to large expansion of CCUG repeats in tissue of individuals with DM2. A definitive evidence would be

the CLIP-RNA Seq of rbFOX1 in skeletal muscle tissues of DM2 patients, but we do not have access to the quantity of human biological material required for such assays. The text is now clarified.

4. Can MBNL1 displace rbFOX1? Do other rbFOX proteins have the same ability in competing with MBNL1 when overexpressed?

Novel UV-crosslinking assays show that addition of increasing amount of rbFOX1 competes with the binding of MBNL1 to CCUG repeats. These results indicate that the binding of MBNL1 and rbFOX1 to CCUG repeats is mutually exclusive, at least in vitro.

Next, we added novel data showing that, alike rbFOX1, addition of rbFOX2 competes MBNL1 binding to CCUG repeats. This is consistent with rbFOX1 and rbFOX2 having an identical RRM recognizing identical RNA sequences. These novel results are presented as Figure 5B and supplementary figure 5A.

5. The authors assume that the binding mechanism of rbFOXs and MBNL1 to CCUG RNA is mechanistically similar to each other. If so, why would the interaction of MBNL1 to CCUG result in sequestration, while the rbFOX/ CCUG RNA interaction doesn't cause rbFOX immobilization?

The mechanisms underlying sequestration of MBNL1 within CCUG RNA foci, while rbFOX1 is mobile, are unclear. A likely explanation would be that the amount of rbFOX1 and rbFOX2 expressed in skeletal muscle simply overcomes the amount of pathogenic RNA containing expanded CCUG repeats. In that aspect, gel shift assays indicate that rbFOX1 binds with a lesser binding affinity to CCUG repeats compared to its normal UGCAUGY RNA target (novel supplementary figure 2). Thus, it is possible that there is a sufficient amount of rbFOX1 in skeletal muscle tissue to bind to both CCUG repeats with a weak affinity and to its normal mRNA targets with a higher affinity. In contrast, MBNL1 presents higher affinity for CCUG repeats compared to its normal pre-mRNA targets (novel supplementary figure 2). Thus, it is likely that expanded CCUG repeats titrate MBNL1 away from binding to its normal mRNA targets.

As an alternative hypothesis, it is possible that the different RNA binding architecture between rbFOX1 and MBNL1 plays a role in their sequestration. Notably, rbFOX1 assembles in large protein complex and form hydrogel (Damianov et al., 2016; Ying et al., 2017), and photoconversion experiments indicate that rbFOX1 moves more slowly than MBNL1. This difference in mobility may reflect different mechanisms of RNA binding that may modulate rbFOX1 and MBNL1 sensibility to sequestration within RNA foci. These different hypotheses are now cited in the discussion.

6. Does the increased level of rbFOX1 expression in C2C12 cells result in a more intense CCUG foci co-localization, and at the same time reduce the intensity of foci-sequestered MBNL1?

Indeed, overexpression of rbFOX1 in DM2 muscle cells reduces co-localization of MBNL1 within CCUG RNA foci, but increases the amount of free diffuse nucleoplasmic MBNL1 (Figure 5C).

Furthermore, we added novel data indicating that shRNA-mediated depletion of endogenous rbFOX1 increases the co-localization of endogenous MBNL1 within CCUG RNA foci and decreases the quantity of free MBNL1. As controls, rbFOX1 overexpression or depletion

have no effect on the localization of MBNL1 within CUG RNA foci in DM1 muscle cells. These novel results are shown as Figures 5C and 5D and supplementary figures 5B and 5C.

7. What is the binding affinity of rbFOX1 to CCUG 20X?

We did not test 20 repeats but the apparent KD of rbFOX1 for 10, 30 and 100 CCUG repeats are 9.7 +/- 1.4 nM (10 CCUG); 6.3 +/- 1.8 nM (30 CCUG); and 3.2 +/- 1.3 nM (100 CCUG) These novel gel shift assays with their apparent kD are now indicated in the supplementary figure 2.

8. In Fig. 2A, CCUG 10X was defined as expanded CCUG repeat. 10 repeats is not considered as expanded. The authors used CCUG 10x and 30x RNAs in the different gel shift experiments. Did they observe CCUG length-dependent binding of rbFOX1?

Indeed, ten CTG or CCTG repeats is not an expansion and the text has been corrected accordingly. Of interest, novel gel-shift assays using 10x, 30x and 100x CCUG repeat indicate that the binding of rbFOX1 to CCUG repeats increases with the number of repeats. These novel results are presented in the supplementary figure 2.

9. Page 7 line 197: "All other tested candidate proteins do not co-localize significantly with CUG or CCUG RNA foci (Figure 1F and Supplemental figure 1)." Would these proteins interact with shorter repeat RNA sequences, e.g. in a gel-shift assay?

As these proteins were not co-localizing within CCUG or CUG RNA foci, we did not investigate them further.

10. What is the rbFOX1 ortholog in Drosophila? What are sequence similarities between the 3 human rbFOX proteins and the Drosophila ortholog? Should a displacement hypothesis be tested in Drosophila, it is more appropriate to use the Drosophila rbFOX1 protein in the overexpression experiment. This is because the MBNL1 counterpart would be endogenous fly MBNL1.

There is only one *Drosophila* ortholog of rbFOX1, 2 and 3, the CG32062 gene. The Drosophila protein shows similarities but also important differences with its human counterparts:

Sequence alignment indicates that the RRM domain of CG32062 is 90% identical to the RNA recognition motif of human or mouse rbFOX1 (Kuroyanagi. 2009). Thus, it is possible that CG32062 may also bind to UGCAUGY RNA sequences, however this has not yet been experimentally demonstrated. In contrast, CG32062 is reported to bind to DNA (Usha and Shashidhara, 2010) and to regulate transcription through association with transcriptional factors (Shukla et al., 2017).

Furthermore, domains of the protein outside the RRM are much more divergent. In particular, CG32062 encodes a 105 kDa protein containing two polyglutamine stretches, while mammalian rbFOX proteins are smaller (~35 to 40 kDa) and lack of any polyglutamine sequences.

Finally, the expression pattern is also different. rbFOX1 and rbFOX2 are expressed in adult mammalian skeletal muscles while CG32062 is absent from adult fly muscles. During embryogenesis, CG32062 is expressed in imaginal discs, in the central and peripheral central nervous system, in the germ bands and in the pericardial cells. Knockdown

experiments indicate that CG32062 is important for development, notably in early oogenesis, specification of the vein-intervein regions, nervous system development and specification of the neuronal sensory organs (Tastan et al., 2010; Usha and Shashidhara, 2010; Shukla et al., 2017). In adult flies, CG32062 is exclusively expressed in the brain and its reduced expression impairs olfactory memory (Güven-Osman et al., 2016).

Thus, as CG32062 has not been formally demonstrated as a functional ortholog of rbFOX1 and is not normally expressed in fly adult skeletal muscles, we have not tested its overexpression in our models of myotonic dystrophies. Of interest, the absence of expression of a *Drosophila* ortholog of rbFOX1 in skeletal muscle may potentially explain the lack of protection against pathogenic CCUG expanded repeats in DM2 flies. This information is now included in the discussion section.

11. Page 13 lines 425-426 “Importantly, expression of rbFOX1 fully corrects the splicing mis-regulations of Fhos and Serca mRNAs caused by expression of expanded CCUG repeats in DM2 flies (Figures 6E and 6F).” Given the RNA-binding property of rbFOX1, this correction may be achieved through mechanism(s) together with/other than the MBNL1 displacement hypothesis. The authors should further explore this.

Additional experiments showing no effect of rbFOX1 overexpression on Fhos and Serca splicing in *Drosophila* have been added as supplementary figures 7C and 7D.

12. To substantiate the competition/displacement hypothesis, it should further be tested in cell-free system(s) using expanded repeat sequences that correspond to the cell/animal models.

We tried to *in vitro* transcribe expanded CCUG repeats (10x, 30x, 100x, 500x, 1000x), but the quality of the radioactively labeled RNA for the largest expansion (500x and 1000x repeats) was not satisfactory, with too many abortive transcriptions, for gel-shift assays.

Minor comments:

Page 9 line 267: It is not clear how many pairs of DM1/DM2 patient samples were presented in this manuscript.

Muscle biopsies and primary human myoblast cells are originating from muscle needle biopsies of six non-DM control individuals, seven individuals with DM1 and seven individuals with DM2. This information is now added to the material and method section.

Page 11 lines 328-329: “These results indicate that, in contrast to MBNL1, the splicing regulatory functions of the rbFOX proteins are not altered by expression of expanded CCUG repeats.” Could this be due to redundancy of rbFOX proteins?

Indeed, it is possible that the amount of rbFOX1 and rbFOX2 in skeletal muscle overcomes the quantity of toxic CCUG RNA repeats. This may be helped by the lesser affinity of rbFOX1 for CCUG repeats compared to its natural mRNA targets or the UGCAUGY RNA motif. In contrast, the higher affinity of MBNL1 for expanded CCUG repeats compared to its normal mRNA targets would favor a titration of MBNL proteins by the expanded CCUG repeats. Gel shifts presenting these novel results are added in supplementary figure 2. Furthermore, these different hypotheses are now presented in the discussion.

Fig. 5B: A mutant rbFOX protein control would be needed to demonstrate the effect of rbFOX altering the subnuclear distribution of MBNL1.

As a control, we added novel experiments showing that shRNA-mediated depletion of endogenous rbFOX1 increases the co-localization of endogenous MBNL1 within CCUG RNA foci and decreases the quantity of free nucleoplasmic MBNL1. In contrast, depletion of rbFOX1 has no effect on the localization of MBNL1 within CUG RNA foci in DM1 muscle cells. These novel data are presented in the supplementary figures 5B and 5C.

Page 9 line 269: Figure 3C was mislabeled as 3H in figure legend.

Page 10 line 313: We “concluded” that rbFOX1 ...

Page 10 line 322: A rbFOX1 knockdown control was missing.

Page 11 line 326: the “extent” of splicing ...

1 000/1,000/1000 are used interchangeably in the manuscript.

These errors are now corrected, and we thank the Referee for his/her help.

REVIEWERS' COMMENTS:

Reviewer #1 (Remarks to the Author):

In this submission new data was included, which raised a number of important questions. The new experimental data collected and the changes introduced in the revised version have significantly strengthened the manuscript.

I would only suggest to simplify description of many figures (especially Supplementary Figs). For example Fig. S2: (i) why C is shown if in A we can see the same experiments or (ii) K and L differ only in rbFOX1 or rbFOX2 used (L should be for example: as in K but for rbFOX2). This should simplify reading of the manuscript.

Reviewer #2 (Remarks to the Author):

The authors have used many complementary methods to demonstrate competition of RBFOXs and MBNLs for CCUGexp but not CUGexp.

Although they were not able to do CLIP experiments due to insufficient material, I feel that the authors have most of reviewer concerns. My one suggestion is given below:

1) Figures 6A and supplementary figure 6B do not show a convincing switch in Clcn1 splicing for CCUG condition with increasing Rbfox1 levels. Since the authors performed a statistical analysis, I suggest attaching a gel picture from another experiment in the main or supplemental figures or use a different target.

The manuscript will be of good quality for publication after that.

Reviewer #3 (Remarks to the Author):

The authors have thoroughly addressed my original critique. Better explanations are now provided for the mass spec identification of rbFox1 protein as one of the RNA binding proteins that specifically binds CCUG repeats. They also have provided better quality images to show co-localization of rbFox1 with CCUG-repeat containing RNA.

The paper has also been improved by extensive responses and new data provided to the address the comments of other reviewers. I agree with the authors that it is not realistic to expect CLIP-seq experiments given the limited amount of patient material available, although ideally these would be great experiments.

Overall I this paper makes a major contribution to our understanding of differences in phenotype of patients with DM1 vs DM2. The study is very well done.

My only comment is that there are a just a few minor errors in grammar or awkward translations, but overall the manuscript is very well written.

Reviewer #4 (Remarks to the Author):

The authors have satisfactorily addressed my concerns.

Minor comments:

Line 215: "rBFOX1" should be "rbFOX1".

Line 547: although "inconsistently"?

Response to Reviewers' comments:

REVIEWER #1:

In this submission, new data was included. The new experimental data collected and the changes introduced in the revised version have significantly strengthened the manuscript. I would only suggest to simplify description of many figures (especially Supplementary Figs). For example: Fig. S2: (i) why C is shown if in A we can see the same experiments or (ii) K and L differ only in rbFOX1 or rbFOX2 used (L should be for example: as in K but for rbFOX2). This should simplify reading of the manuscript.

Indeed, some of the figure legends were redundant and have been simplified. The text is now hopefully clearer and easier to read.

REVIEWER #2:

The authors have used many complementary methods to demonstrate competition of RBFOXs and MBNLs for CCUGexp but not CUGexp. Although they were not able to do CLIP experiments due to insufficient material, I feel that the authors have most of reviewer concerns. My one suggestion is:

Figures 6A and supplementary figure 6B do not show a convincing switch in *Clcn1* splicing for CCUG condition with increasing *Rbfox1* levels. Since the authors performed a statistical analysis, I suggest attaching a gel picture from another experiment in the main or supplemental figures or use a different target. The manuscript will be of good quality for publication after that.

Two examples of gels showing *Clcn1* exon 6B splicing correction upon rbFOX1 overexpression are now added presented in supplementary figure 6A.

REVIEWER #3:

The authors have thoroughly addressed my original critique. Better explanations are now provided for the mass spec identification of rbFox1 protein as one of the RNA binding proteins that specifically binds CCUG repeats. They also have provided better quality images to show co-localization of rbFox1 with CCUG-repeat containing RNA. The paper has also been improved by extensive responses and new data provided to the address the comments of other reviewers. I agree with the authors that it is not realistic to expect CLIP-seq experiments given the limited amount of patient material available, although ideally these would be great experiments. Overall, I this paper makes a major contribution to our understanding of differences in phenotype of patients with DM1 vs DM2. The study is very well done. My only comment is that there are a just a few minor errors in grammar or awkward translations, but overall the manuscript is very well written.

The text is now amended.

REVIEWER #4:

The authors have satisfactorily addressed my concerns. Minor comments:

Line 215: "rbFOX1" should be "rbFOX1".

Line 547: although "inconsistently"?

These errors have been corrected, and we would like to thank all four Referees for their help.